# WHEN DO DISTANT DEPENDENCIES MATTER? DIAGNOSTICS FOR LONG-RANGE PROPAGATION IN GNNS

## ABSTRACT

Graph Neural Networks (GNNs) propagate information locally through message passing. While local propagation is often sufficient for short-range tasks, performance can degrade when distant interactions are required. In this paper, we introduce a diagnostic metric that quantifies the role of long-range propagation. The metric is derived from margin-aligned sensitivities, providing an interpretable measure of the dominance of one-hop neighbors in margin-relevant influence. Using this diagnostic, we show that the need for long-range propagation is dataset- and architecture-dependent, rather than universal. We further demonstrate that this diagnostic metric is predictable from well-studied graph-theoretic measures, aligning with the assumptions of rewiring-based approaches. Finally, we show how the diagnostic can be leveraged during training: we design an additional layer that selectively incorporates sensitivity to long-range dependencies and can be applied to any standard GNN backbone. Experiments on both node- and graph-level benchmarks demonstrate consistent gains over rewiring-based methods, without altering the original graph topology.

## 1 INTRODUCTION

Graph Neural Networks (GNNs) (Goller & Kuchler, 1996; Scarselli et al., 2008; Bruna et al., 2014) learn from relational structure by propagating and aggregating messages (Gilmer et al., 2017), achieving strong performance across chemistry, social networks, recommendation, and spatiotemporal forecasting (Zhou et al., 2020; Duval et al., 2023; Sharma et al., 2024; Wu et al., 2022; Castro-Correa et al., 2024). When task-relevant evidence lies many hops away, however, performance can degrade. A leading explanation is over-squashing (Alon & Yahav, 2021; Topping et al., 2022; Di Giovanni et al., 2023): signals routed through sparse or bottlenecked regions are compressed into fixed-size embeddings, diluting information.

Prior work analyzes over-squashing from two angles (Arnaiz-Rodriguez & Errica, 2025). The structural view Alon & Yahav (2021); Topping et al. (2022) attributes failure to graph bottlenecks, e.g., negative curvature (Topping et al., 2022), low effective resistance (Black et al., 2023), and small spectral gaps (Karhadkar et al., 2023) and typically intervenes by rewiring the graph structure (Attali et al., 2024b; Akansha, 2025). The other angle is the computational (i.e., model) bottlenecks perspective, where the limitation stems from message passing itself: finite-depth, locality aggregations restrict the receptive field and progressively compress signals, so information from long-distance nodes is hard to both reach and preserve (Di Giovanni et al., 2023; Arnaiz-Rodriguez & Errica, 2025). These perspectives are complementary, yet it often remains unclear whether accuracy gains after rewiring truly stem from alleviating over-squashing, or when increasing long-range influence is desirable for the task at hand.

We take a task-aligned view and argue that dependence on long-range information is inherently task- and node-specific. Rather than universally amplifying long-range influence, interventions should be adaptive. To ground this claim, we introduce a margin-aligned, Jacobian-based sensitivity index that quantifies, for a trained GNN, how a node's (or graph's) true classification margin responds to one-hop versus multi-hop perturbations (§3). The resulting long-range capture index $p_u \in [0, 1]$ (with graph-level aggregate $\rho_G$) is a bounded, interpretable "share-at-one-hop" measure that directly traces computational limitations of message passing.

Using this diagnostic, we show that GNNs implicitly operate in local and non-local regimes: some nodes behave as if decisions are locally controlled (high $p_u$), others as if non-local inputs dominate (low $p_u$). Importantly, the model's performance co-varies with this sensitivity in a task-dependent manner: across benchmarks and backbones, the margin–sensitivity correlation is approximately linear, but its sign and magnitude vary with dataset and architecture. In some settings, reducing the one-hop share helps; in others, preserving locality is beneficial. This organization, along a single sensitivity axis, explains when long-range propagation aids or harms prediction (§3).

We then uncover a bridge to graph structure: the true sensitivity defined via the margin can be predicted from topology alone (§4.1). A sparse linear model (Lasso) on structural indicators widely used in structural rewiring (e.g., curvature and effective resistance) yields accurate, structure-only proxies for $p_u$ at node and graph levels. This link connects structural accounts of over-squashing to model-level behavior and enables label-free estimation at test time.

Finally, we convert these insights into a minimal intervention at readout. We introduce FLAN (§4.2), a rewiring-free, lightweight long-range layer that conditions the classifier on the structure-predicted proxy $\hat{p}_u$. The layer applies a small translation and a one-parameter diagonal reweighting of the encoder representation, effectively letting the readout adapt across local vs. non-local regimes while keeping the encoder and topology unchanged. Empirically, this plug-in improves accuracy across GNN backbones and datasets (§5), offering a simple and time-efficient alternative to graph rewiring.

The main contributions of this paper are summarized as follows:

1. We introduce a task-aligned, Jacobian-based diagnostic of long-range sensitivity at node ($p_u$) and graph ($\rho_G$) scales.

2. We demonstrate that this diagnostic is accurately predicted from graph structure via a sparse structural model, linking structural bottlenecks to trained model sensitivity.

3. We provide cross-dataset/backbone evidence that margins vary monotonically along the sensitivity axis, with task-dependent sign.

4. Finally, we design FLAN, a rewiring-free, parameter-efficient conditioning layer that leverages the predicted sensitivity $\hat{p}_u$ to improve performance without changing the graph or increasing depth.

Our study contributes to a unified understanding of over-squashing: structural features forecast a trained model's long-range *sensitivity*; errors organize along this sensitivity axis; and an adaptive, low-capacity correction exploits this organization to deliver consistent gains (Arnaiz-Rodríguez & Errica, 2025; Bechler-Speicher et al., 2025).

**Reproducibility.** The source code to reproduce our experiments is available[1].

## 2 BACKGROUND AND RELATED WORK

We start by introducing notations used throughout this paper. Let $G = (V, E$ be a simple, undirected, unweighted graph with node-feature matrix $\mathbf{H} \in \mathbb{R}^{|V| \times d}$. Let $\mathbf{A} \in \{0, 1\}^{|V| \times |V|}$ be its adjacency matrix, $\mathbf{D} = \mathrm{diag}(d_u)_{u \in V}$ the degree matrix, $\mathbf{P} = \mathbf{D}^{-1}\mathbf{A}$ the transition matrix and the normalized Laplacian is $\mathbf{L}_{\mathrm{norm}} = \mathbf{I} - \mathbf{D}^{-1/2}\mathbf{A}\mathbf{D}^{-1/2}$. For $u \in V$, we denote its neighborhood by $\mathcal{N}(u) = \{v \in V : (u, v) \in E\}$.

**Message passing in GNNs.** GNNs are built upon the message passing mechanism, in which node representations are refined through local interactions (Gilmer et al., 2017). At each layer, a node aggregates information from its neighbors using a permutation-invariant function, followed by a learnable transformation. Formally, for a node $i \in \mathcal{V}$, its representation at layer $k + 1$ is defined as:

$$\mathbf{h}_i^{(k+1)} = \phi\left(\mathbf{h}_i^{(k)}, \bigoplus_{j \in \mathcal{N}(i)} \psi(\mathbf{h}_j^{(k)})\right),$$

---

[1]https://anonymous.4open.science/r/FLAN_ICLR_2026-3E65

where $\mathbf{h}_i^{(k)}$ denotes the representation of node $i$ at layer $k$, $\psi$ the message function, and $\phi$ the update function. The operator $\bigoplus$ denotes a permutation-invariant aggregation function such as summation, mean, or maximum. This iterative procedure allows GNNs to integrate both feature and structural information from local neighborhoods. Message passing is effective when task-relevant information is local and can be aggregated within only a few hops, which is typically the case in homophilic graphs (Zhu et al., 2021). For long-range dependencies, communication across a distance $d$ requires $\mathcal{O}(d)$ message-passing layers (Barceló et al., 2020). Increasing the depth in this way amplifies over-squashing (Di Giovanni et al., 2023; Akansha, 2025) and over-smoothing (Rusch et al., 2023; Giraldo et al., 2023).

**Over-squashing, long-range interactions, and graph rewiring.** Over-squashing occurs when information from exponentially large neighborhoods must be compressed into fixed-size node embeddings within a limited number of message-passing layers (Alon & Yahav, 2021; Topping et al., 2022). As the receptive field expands with depth, the aggregation function is forced to encode ever larger amounts of information into a bounded representation, creating a bottleneck that severely limits the ability of GNNs to capture long-range dependencies, particularly in graphs with sparse connectivity or complex topology.

Graph rewiring addresses over-squashing and long-range dependencies by modifying the input topology of a GNN, alleviating structural bottlenecks that hinder the propagation of information across distant nodes. Early work focuses on curvature-based rewiring, adding edges around regions with highly negative discrete curvature that indicate bottlenecks (Topping et al., 2022; Giraldo et al., 2023; Nguyen et al., 2023; Fesser & Weber, 2023). Because discrete curvature measures are inherently local (Forman, 2003; Ollivier, 2007; Samal et al., 2018), subsequent approaches have targeted more global signals, either increasing the spectral gap to improve connectivity and mixing (Banerjee et al., 2022; Karhadkar et al., 2023) or minimizing effective resistance, which models the difficulty of information transmission between node pairs (Black et al., 2023).

More recently, a complementary line of work incorporates node features into the rewiring techniques. For example, Delaunay-based rewiring reconstructs the graph by performing a Delaunay triangulation in feature space, thereby removing edges that exhibit extreme discrete curvature (Attali et al., 2024a; 2025). Other approaches jointly modify the topology and the initial node features to maximize the spectral alignment between the feature signal and the structural information (Linkerhägner et al., 2025). Finally, intra-community rewiring guided by the cosine similarity of node features has been proposed to densify connections among similar nodes while preserving community-level structure (Rubio-Madrigal et al., 2025).

One can distinguish between different types of bottlenecks. Structural bottlenecks arise from the graph's topology (narrow cuts, hubs, or low expansion) that restrict information flow regardless of the model. Computational bottlenecks stem from the message-passing computation itself: even on favorable graphs, signals and gradients from distant nodes attenuate through repeated local updates. Most existing metrics target structural limits; far fewer directly capture the computational one. The computational bottleneck is often studied via Jacobians : Topping et al. (2022); Di Giovanni et al. (2023) show that node-to-node sensitivity decays exponentially with graph distance, explaining the difficulty of propagating long-range information in GNNs.

## 3 GNN PERFORMANCE AND LONG-RANGE DEPENDENCIES

In this section, we extend the study of long-range effects and over-squashing by grounding the analysis in the model's Jacobian (Topping et al., 2022; Di Giovanni et al., 2023; Giovanni et al., 2024). Rather than focusing on pairwise dependencies between individual nodes, we directly quantify both the distance (in graph terms) and the amount of task-relevant information that a node's representation can capture in a classification task. Concretely, we aggregate margin-aligned Jacobian sensitivities into a one-hop dominance measure, quantifying how much of the margin-relevant signal is captured locally rather than over longer ranges. We then examine how this long-range signal relates to architectural performance. Importantly, instead of relying solely on accuracy, we evaluate with the classification margin, which provides a finer view of confidence and decision robustness. This margin-aligned perspective allows us to connect distance-structured sensitivity to accuracy gains offering a clear diagnostic of when and how architectures benefit from long-range information. Below, we elaborate on the different steps.

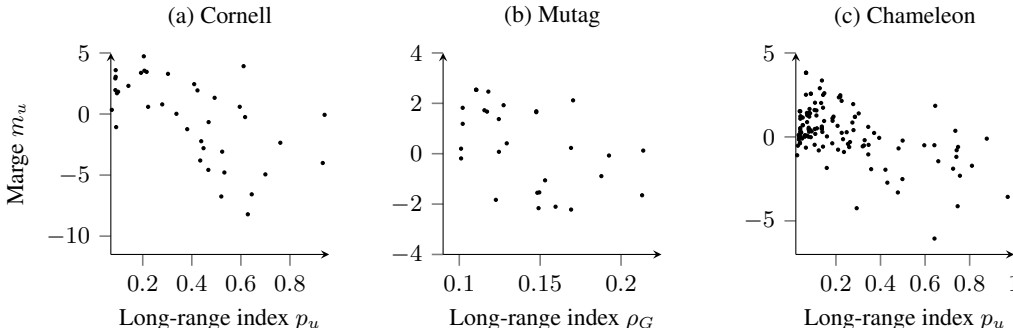

Figure 1: Correlation between classification margin and the long-range capture index $\bar{\rho}_u$: (a) COR-NELL (node classification), (b) MUTAG (graph classification), (c) CHAMELEON (node classification). Higher $\bar{\rho}_u$ values (dominant 1-hop contribution) tend to coincide with smaller margins when long-range evidence is required.

**Task-aware node margin.** For node classification, let $\mathbf{z}_u \in \mathbb{R}^C$ be the logits predicted for node $u \in V$ with ground-truth label $y_u \in \{1, \ldots, C\}$. The node-level margin is defined as

$$m_u = \mathbf{z}_u[y_u] - \max_{c \neq y_u} \mathbf{z}_u[c]. \tag{1}$$

The margin is directly aligned with the downstream task: $m_u > 0$ indicates correct classification; larger values reflect a larger separation from the closest competing class. For graph classification, we similarly define a graph-level margin $m_G$. Specifically, letting $\mathbf{z}_G \in \mathbb{R}^C$ denote pooled graph logits with label $y_G$, we set $m_G = \mathbf{z}_G[y_G] - \max_{c \neq y_G} \mathbf{z}_G[c]$.

**Label-aware sensitivity.** To attribute the classification margin to input features, we compute the magnitude of the first-order effect:

$$J_{s,g}^u := \left| \frac{\partial m_u}{\partial \mathbf{H}_{s,g}^{(0)}} \right|, \tag{2}$$

where $s \in V$ indexes a source node and $g \in \{1, \ldots, F\}$ a feature dimension. Intuitively, $J_{s,g}^u$ measures how much the classification margin of node $u$ changes in response to a small change in feature $g$ of source node $s$. For graph classification, we analogously define $J_{s,g}^G := \left| \partial m_G / \partial \mathbf{H}_{s,g}^{(0)} \right|$.

**Distance-binned aggregation.** Having computed the label-aware sensitivities, we next aggregate them according to graph distance from a reference node $u$:

$$S_{u,g}(k) = \sum_{s:\, \mathbf{D}(s,u)=k} J_{s,g}^u, \qquad k = 0, 1, 2, \ldots, \tag{3}$$

with $\mathbf{D}(\cdot, \cdot)$ the number-of-hops on the input graph. This yields a distance-resolved profile of label-aware influence; in message passing GNNs, contributions beyond the network depth are typically negligible, but we retain the full histogram for completeness. For graph classification, we use the same binning around $u$: $S_{u,g}^G(k) := \sum_{s:\, \mathbf{D}(s,u)=k} J_{s,g}^G$.

**Long-range capture index.** We quantify the fraction captured only by the one-hop neighborhood; for a node $u$ we define:

$$\rho_{u,g} = \frac{S_{u,g}(1)}{\sum_{k \geq 1} S_{u,g}(k)} \in [0, 1]. \tag{4}$$

Normalizing by $\sum_{k \geq 1}$ makes $\rho_{u,g}$ scale-invariant to global rescalings of gradients. For graph classification, this is defined as $\rho_{u,g}^G := \frac{S_{u,g}^G(1)}{\sum_{k \geq 1} S_{u,g}^G(k)} \in [0, 1]$.

We obtain a node- and a graph-level score by averaging over features as follows:

| **Node-level index** | **Graph-level index** |
|---|---|

$$p_u = \frac{1}{F} \sum_{g=1}^F \rho_{u,g} \in [0,1]. \quad (5) \qquad\qquad \rho_G = \frac{1}{|V| F} \sum_{u \in V} \sum_{g=1}^F \rho_{u,g}^G \in [0,1]. \quad (6)$$

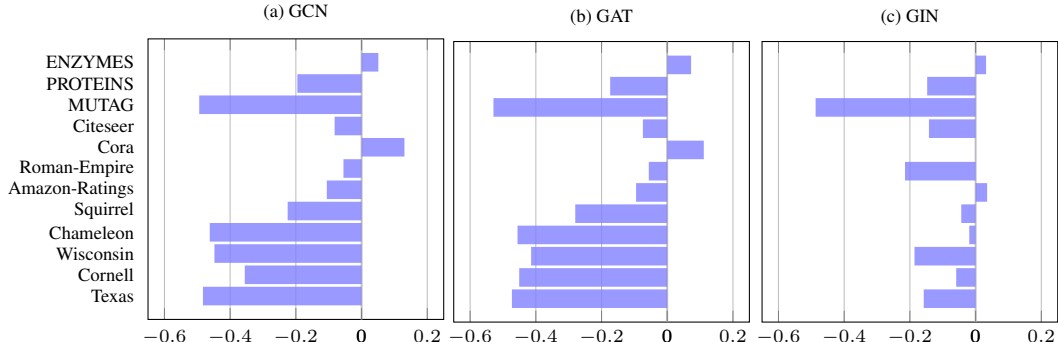

Figure 2: Correlation (mean over 20 runs) between the classification margin and the long-range capture index $\rho_u$ for three backbones (GCN, GAT, GIN). Negative values indicate that performance increases as the 1-hop share decreases, i.e., when long-range propagation becomes more informative.

Larger $p_u$ (and $\rho_G$) indicates that margin-relevant influence is disproportionately concentrated at distance 1, indicating limited long-range transmission to $u$. Equivalently, this long-range diagnostic index can be interpreted as a one-hop dominance score for node $u$: it summarizes how much of the margin-aligned sensitivity that reaches $u$ is already captured in its immediate neighborhood as opposed to arriving from longer ranges. Let us note here that, although the range measure proposed by Bamberger et al. (2025) also leverages Jacobian information, it is designed to be task-agnostic and quantifies how far Jacobian/Hessian influence can propagate. In contrast, our diagnostic is margin-aligned, indicating when distant information helps or hurts the decision boundary.

**How does the long-range capture index relate to the classification task?** To analyze GNN's behavior on a given graph dataset, we study the correlation between the classification margin $m_u$ (Eq. (1)) and the long-range capture index (Eq. (5) and (6)). Figure 1 illustrates the trends on Chameleon, Cornell, and MUTAG. Figure 2 reports the mean correlation over 20 runs for GCN (Kipf & Welling, 2017), GAT (Veličković et al., 2018), and GIN (Xu et al., 2019) across nine node and three graph classification datasets commonly used in graph rewiring experiments (Topping et al., 2022; Giraldo et al., 2023; Attali et al., 2024a; Karhadkar et al., 2023; Nguyen et al., 2023; Liang et al., 2025). Experimental details are provided in Appendix A.1.

Across datasets, the correlation between the classification margin and the long-range capture index is not universal but depends on both the dataset and the GNN backbone. On heterophilic graphs, GCN and GAT exhibit negative correlations, indicating that margins improve as reliance on one-hop information decreases, i.e., long-range capture helps. On homophilic graphs, the correlation is close to zero and slightly positive, indicating that one-hop information is more informative for the task than long-range information, which aligns with the structure of the graph. GCN and GAT exhibit broadly similar behavior on node classification datasets: their diffusion-based aggregation yields greater variability in the one-hop share $p_u$. In contrast, GIN operates in a distinct regime: its sum aggregation followed by an MLP favors local evidence, yielding larger and more tightly concentrated $p_u$ and a reduced reliance on long-range contributions. Our findings are not specific to 1-hop choice in Eq. (4): enlarging the "short-range" bin (e.g., to 1–2 or 1–3 hops) changes index magnitude but leaves its correlation with the margin $m_u$ essentially unchanged (see Appendix A.2).

## 4 FROM DIAGNOSTICS TO LONG-RANGE INTERVENTION

### 4.1 DECODING LONG-RANGE EFFECTS FROM GRAPH TOPOLOGY

To mitigate long-range dependencies, rewiring methods typically rely on structural measures. In this section, we ask whether topology alone can explain and predict the node-wise long-range capture index, i.e., whether the structural indicators used for rewiring recover $p_u$ or $\rho_G$. To obtain an interpretable link between graph topology and our diagnostic index, we estimate a sparse linear relation whose coefficients identify the indicators that affect $p_u$ (or $\rho_G$) along with the sign and magnitude of their effects. To this end, we use four measures that are widely used in graph rewiring methods.

| Dataset | GCN | GAT | GIN |
|---|---|---|---|
| Texas | $0.6377_{\pm 0.10}$ | $0.3859_{\pm 0.14}$ | $0.5127_{\pm 0.10}$ |
| Cornell | $0.7037_{\pm 0.15}$ | $0.5161_{\pm 0.18}$ | $0.5560_{\pm 0.11}$ |
| Wisconsin | $0.5653_{\pm 0.11}$ | $0.4269_{\pm 0.12}$ | $0.5453_{\pm 0.09}$ |
| Chameleon | $0.4270_{\pm 0.05}$ | $0.3509_{\pm 0.04}$ | $0.6769_{\pm 0.28}$ |
| Squirrel | $0.4349_{\pm 0.01}$ | $0.3258_{\pm 0.19}$ | $0.4349_{\pm 0.01}$ |
| Amazon-Ratings | $0.7897_{\pm 0.01}$ | $0.4000_{\pm 0.04}$ | $0.8055_{\pm 0.02}$ |
| Roman-empire | $0.6831_{\pm 0.03}$ | $0.6551_{\pm 0.03}$ | $0.4070_{\pm 0.04}$ |
| Cora | $0.3050_{\pm 0.02}$ | $0.3600_{\pm 0.02}$ | $0.2737_{\pm 0.03}$ |
| Citeseer | $0.3100_{\pm 0.03}$ | $0.3644_{\pm 0.04}$ | $0.3377_{\pm 0.04}$ |
| MUTAG | $0.9922_{\pm 0.00}$ | $0.9751_{\pm 0.03}$ | $0.9867_{\pm 0.01}$ |
| PROTEINS | $0.9560_{\pm 0.01}$ | $0.9564_{\pm 0.01}$ | $0.9531_{\pm 0.01}$ |
| ENZYMES | $0.7548_{\pm 0.13}$ | $0.7548_{\pm 0.13}$ | $0.7862_{\pm 0.09}$ |
| IMDB | $0.8340_{\pm 0.01}$ | $0.8349_{\pm 0.01}$ | $0.7567_{\pm 0.02}$ |

Table 1: $R^2$ mean on the test set of Lasso regression using structure indicators to predict the capture index across different backbones and datasets.

**(i) PageRank (Page et al., 1999).** PageRank is a random-walk centrality that highlights highly influential nodes. It is used in GNNs to guide rewiring or capacity allocation via higher-order diffusion (Klicpera et al., 2019), central virtual nodes (Qian et al., 2024; Southern et al., 2025), or node-wise capacity scaling (Choi et al., 2024). Formally, $\boldsymbol{\pi}^\top = (1-\alpha)\, \mathbf{1}^\top/|V| + \alpha\, \boldsymbol{\pi}^\top \mathbf{D}^{-1}\mathbf{A}$ with $\alpha \in (0,1)$.

**(ii) Forman–Ricci edge curvature (Samal et al., 2018).** Edges with highly negative curvature typically coincide with structural bottlenecks that intensify over-squashing (Alon & Yahav, 2021; Topping et al., 2022), whereas edges with highly positive curvature promote intra-cluster propagation and can accentuate over-smoothing (Nguyen et al., 2023). These curvature signals motivate curvature-aware rewiring that targets bottlenecks to improve information flow (Topping et al., 2022; Giraldo et al., 2023; Nguyen et al., 2023; Fesser & Weber, 2023; Liu et al., 2023). For an edge $e = (u,v)$, we use the augmented Forman curvature $F(u,v) = 4 - (d_u + d_v) + 3\,t_{uv}$, where $t_{uv}$ is the number of triangles incident to $(u,v)$. Let $q_{0.1}$ and $q_{0.9}$ denote the 10th and 90th percentiles of $\{F(e)\}_{e \in E}$. To obtain node-level indicators, for each node $u$ we count incident edges in the bottom and top deciles: $F_{10}(u) = |\{v \in N(u) : F(u,v) \leq q_{0.1}\}|$, $F_{90} = |\{v \in N(u) : F(u,v) \geq q_{0.9}\}|$. A large $\mathtt{bot}_{0.1}(u)$ signals exposure to strongly negative-curvature (bottleneck) edges, while a large $\mathtt{top}_{0.9}(u)$ characterizes cohesive, intra-cluster ties.

**(iii) Mean commute time.** Commute time quantifies the difficulty of long-range transmission, large values highlight regions where propagation is inefficient and motivate rewiring to improve long range connectivity (Di Giovanni et al., 2023; Black et al., 2023; Barbero et al., 2024; Sterner et al., 2024; Zhuo et al., 2025). Formally we define the mean commute time as $C_{uv} = 2\,|E|\,R_{uv}$, where $R_{uv}$ is the effective resistance (Chandra et al., 1989) between node $u$ and $v$. For a node $u$ the mean commute time is defined as $\overline{C}(u) = \frac{1}{|V|-1}\sum_{j \in V \setminus \{u\}} C_{uj}$. Large $\overline{C}(u)$ indicates costly long-range access between $u$ and the rest of the graph (Di Giovanni et al., 2023).

Finally, the node-level structural indicator is the aggregation of four measures:

$$\mathbf{S}(u) \;=\; \big[\overline{C}(u),\; \pi(u),\; F_{10}(u),\; F_{90}(u)\big] \in \mathbb{R}^4.$$

**Sparse linear model for long-range capture index.** Let $\mathbf{S} \in \mathbb{R}^{N \times 4}$ stack $\boldsymbol{s}(u)$ over nodes. We fit a sparse linear predictor of the task-aligned index $p_u$ or $\rho_G$:

$$(\widehat{\beta}_0, \widehat{\boldsymbol{\beta}}) \;\in\; \arg\min_{\beta_0, \boldsymbol{\beta}} \; \frac{1}{2|\mathcal{I}_{\text{train}}|} \sum_{u \in \mathcal{I}_{\text{train}}} \big(p_u - \beta_0 - \mathbf{S}_u^\top \boldsymbol{\beta}\big)^2 \;+\; \lambda \|\boldsymbol{\beta}\|_1, \tag{7}$$

with $\lambda$ chosen by $K$-fold cross-validation on training nodes. We report the test $R^2(\hat{p}_u, p_u)$ in Table 1.

**Can structure alone predict the long-range capture index?** On graph classification, the structure-only proxy closely matches the model-derived $p_u$. For node classification, the alignment is strongest on heterophilous datasets and attenuates on homophilous ones, where one-hop evidence

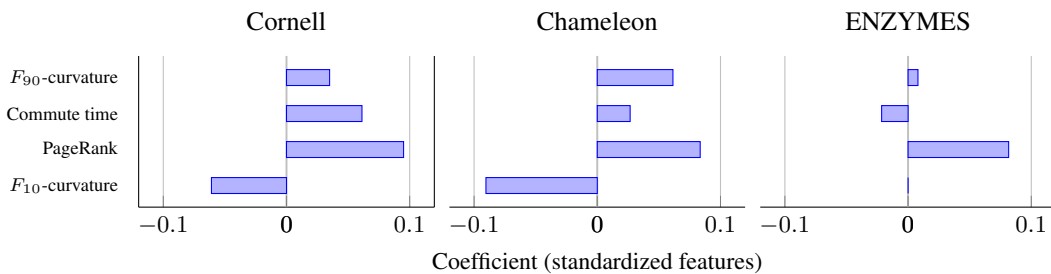

Figure 3: Lasso node indicators influencing the long-range capture index.

dominates. These trends hold across GCN, GAT, and GIN, indicating robustness to the backbone. They further confirm that $p_u$ and $\rho_u$ are largely topology-driven, reflecting the same structural signals that rewiring methods leverage. To improve interpretability, we complement Table 1 with two additional baselines based on node degree ($\mathbf{S}(u) = [\min \deg(u), \text{mean} \deg(u), \max \deg(u)]$), as well as a random baseline obtained by shuffling the targets. As shown in Appendix A.2 (Table ), such local degree statistics are not sufficient to predict the model-derived long-range index $p_u$, whereas the structural measures used in graph rewiring methods achieve substantially higher test $R^2$.

**Analysis of the Lasso coefficients.** Figure 3 reports the Lasso coefficients on three datasets. The coefficients vary across datasets, indicating that different structural indicators modulate the long-range capture index. For instance, a lower mean commute time corresponds to a slight increase in $p_u$ on Cornell, whereas it correlates negatively with $\rho_G$ on ENZYMES. We also observe that higher PageRank, i.e., greater centrality, typically coincides with a high $p_u$, suggesting that highly central nodes struggle to capture long-range information; their capacity concentrates on strong one-hop signals and thereby reduces the contribution of distant nodes, in line with Choi et al. (2024).

**Topological bottlenecks and long-range propagation.** For some datasets we observe that incidence to highly negative Forman–curvature edges is negatively associated with $p_u$. While such edges are often labeled as bottlenecks, negatively curved edges can also act as bridges linking distinct communities: being incident to one effectively grants a node access to many distant neighbors, which lowers $p_u$. This result corroborates the observation of Arnaiz-Rodríguez & Errica (2025) that not all bottlenecks are harmful to long-range dependence; some enable controlled long-range reach.

**Can structural properties predict the true classification margin?** We replaced the sparse linear model of (7) used to predict the long-range index $p_u$ from our four node-level structural indicators with an otherwise identical Lasso that instead regresses the true node margin $m_u$ from the same features. On held-out nodes across datasets and backbones, this topology-only regression of $m_u$ yielded very low $R^2 (\approx 0)$, in contrast to the substantially higher $R^2$ obtained when predicting $p_u$. This outcome is consistent with our framework: $m_u$ is jointly determined by labels, node features, and the learned encoder, and its association with long-range effects even changes sign across datasets, whereas $p_u$ isolates a one-hop-dominance property that is largely structural and thus predictable from these indicators. In short, topology helps locate where long-range pressure exists, but it cannot by itself reconstruct how confident the model is in a class decision.

### 4.2 FLAN: A Rewiring-free Long-range Layer

Our analysis shows that node margins vary systematically with $p_u$ (§3); and that $p_u$ is predictable from structure alone (§4.1). A single global linear head must therefore compromise across local vs. non-local regimes. We propose a topology-preserving readout adjustment whose per-node intensity is driven by the measurable diagnostic $\hat{p}_u$, without changing the graph or increasing depth (main results and ablation in §5).

**Setup.** Let $\Phi_{\boldsymbol{\theta}}$ be a frozen GNN encoder with $L$ layers and let $\mathbf{h}_u^{(L)} = \Phi_{\boldsymbol{\theta}}(\cdot)_u \in \mathbb{R}^d$ be the embedding of node (or graph) $u$. Let $p_u \in [0, 1]$ denote the long-range capture index in (5); we estimate it using the sparse linear model of (7) over structural indicators, yielding $\hat{p}_u \in [0, 1]$.

**FLAN Layer.** We attach a gating map $g_\phi : \mathbb{R}^d \times \mathbb{R} \to \mathbb{R}^d$ with parameters $\phi = (\boldsymbol{w}_\gamma, \boldsymbol{w}_\beta)$, $\boldsymbol{w}_\gamma, \boldsymbol{w}_\beta \in \mathbb{R}^d$:

$$\mathbf{z}_u = \sigma_s(\boldsymbol{w}_\gamma \, \hat{p}_u) \odot \mathbf{h}_u^{(L)} + \boldsymbol{w}_\beta \, \hat{p}_u, \tag{8}$$

where $\sigma_s(\cdot)$ is an elementwise sigmoid and $\odot$ is the Hadamard product. The classifier is linear,

$$\mathrm{logits}(u) = \mathbf{W}\mathbf{z}_u + b, \qquad \mathbf{W} \in \mathbb{R}^{C \times d}, \; \mathbf{b} \in \mathbb{R}^C. \tag{9}$$

During training, we optimize $(\phi, \mathbf{W}, \mathbf{b})$ with cross-entropy; $\boldsymbol{\theta}$ is kept fixed. Intuitively, FLAN uses the diagnostic signal $\hat{p}_u$ to apply a per-node rescaling of $\mathbf{h}_u^{(L)}$ and a per-node bias shift in logit space.

**Geometric view.** The additive term $\boldsymbol{w}_\beta \hat{p}_u$ implements a $p$-dependent *translation* of the decision boundary (a family of parallel hyperplanes indexed by $\hat{p}$). The multiplicative term $\sigma_s(\boldsymbol{w}_\gamma \hat{p}_u) \odot \mathbf{h}_u^{(L)}$ implements a $p$-dependent *reweighting* of coordinates, effectively tilting the separator. The sensitivity index compresses long-range demand into a single axis that is highly predictive of where the baseline fails. Because the dominant error varies monotonically with $p_u$, this rank-1 translation plus diagonal reweighting is a minimal intervention that corrects the under-performing $p$ regime.

## 5 EXPERIMENTS

To evaluate the effect of the proposed FLAN layer, we evaluate it on node classification tasks spanning both homophilic graphs (Sen et al., 2008) and heterophilic graphs (Rozemberczki et al., 2021; Tang et al., 2009), as well as on graph classification benchmarks (Morris et al., 2020). The latter are widely adopted in the evaluation of rewiring methods, since their structures are tightly coupled to the downstream task and require the propagation of long-range dependencies (Karhadkar et al., 2023). Additional results on long-range benchmark datasets are provided in the Appendix B .

**Baseline models.** We compare FLAN to seven state-of-the-art rewiring techniques: the curvature-based methods SDRF (Topping et al., 2022) and BORF (Nguyen et al., 2023); the spectral rewiring method FoSR (Karhadkar et al., 2023); the resistance-based approach GTR (Black et al., 2023); LASER (Barbero et al., 2024) a Random Walk Rewiring Based method; DR (Attali et al., 2024a) leverages node features to perform Delaunay triangulation-based rewiring; GOKU (Liang et al., 2025), two-stage densify–then-sparsify rewiring that preserves spectral properties and improves long-range information flow.

**Experimental setup.** We follow the evaluation protocol of (Liang et al., 2025): GNN hyperparameters are fixed across methods (learning rate $1\mathrm{e}{-}3$, hidden dimension 64, 4 layers), while rewiring hyperparameters are tuned per method. Baseline results are reported from (Liang et al., 2025).

**Results.** Table 2 reports the results of and graph classification tasks across different GNN backbones. Overall, without altering the input topology, FLAN improves backbone GNN performance by more than 12% on average, and it outperforms recent rewiring baselines. On graph classification, it outperforms all rewiring methods with both GCN and GIN backbones; this is consistent with the higher and more stable $R^2$ of the structure-only proxy for $\hat{p}_G$, which makes the scalar conditioning particularly effective at the graph level. On node classification, the FLAN layer is competitive but not always state-of-the-art on small heterophilic datasets, where the correlation between margin and $p_u$ (and the corresponding $R^2$) exhibits high variance, making gains less stable. We also observe benefits on homophilous datasets, where one-hop evidence dominates and the layer acts conservatively rather than over-correcting. Figure 4 shows that the gains arise not from added capacity, but from the long-range signal encoded by the predicted long-range index.

**Time comparison.** In Appendix 14, we compare FLAN's preprocessing runtime against graph–rewiring baselines. The reported times include (i) Jacobian–margin evaluation, (ii) computation of structural indicators, and (iii) Lasso fitting for $\hat{p}_G$. On average, our method is $10^1$–$10^3 \times$ faster than curvature-based rewiring (Topping et al., 2022; Nguyen et al., 2023), spectral-gap-based rewiring (Karhadkar et al., 2023), and resistance-based rewiring (GTR) (Black et al., 2023).

**Ablation studies.** To confirm that improvements are diagnostic-driven, Figure 4 compares $\hat{p}$-conditioning to shuffled $\hat{p}$ (permuted across graphs) and to a margin-conditioned scalar. Only FLAN yields significant gains over the backbone GCN, supporting that the benefits arise from the structure-predicted index rather than added capacity or margin tuning. In Appendix C, we further analyze the

(a) Node classification (Backbone: GCN)

| Method | Cora | Citeseer | Texas | Cornell | Wisconsin | Chameleon |
|---|---|---|---|---|---|---|
| None | $86.7_{\pm0.3}$ | $72.3_{\pm0.3}$ | $44.2_{\pm1.5}$ | $41.5_{\pm1.8}$ | $44.6_{\pm1.4}$ | $59.2_{\pm0.6}$ |
| SDRF | $86.3_{\pm0.3}$ | $72.6_{\pm0.3}$ | $43.9_{\pm1.6}$ | $42.2_{\pm1.5}$ | $46.2_{\pm1.2}$ | $59.4_{\pm0.5}$ |
| FOSR | $85.9_{\pm0.3}$ | $72.3_{\pm0.3}$ | $46.0_{\pm1.2}$ | $40.2_{\pm1.1}$ | $48.3_{\pm1.3}$ | $59.3_{\pm0.6}$ |
| BORF | $87.5_{\pm0.2}$ | $73.8_{\pm0.2}$ | $49.4_{\pm1.8}$ | $50.8_{\pm1.1}$ | $50.3_{\pm0.9}$ | $61.5_{\pm0.4}$ |
| DR | $78.4_{\pm1.2}$ | $69.5_{\pm1.6}$ | $67.8_{\pm2.5}$ | $57.8_{\pm1.9}$ | $62.8_{\pm2.1}$ | $58.6_{\pm0.8}$ |
| GTR | $87.3_{\pm0.4}$ | $72.4_{\pm0.3}$ | $45.9_{\pm1.9}$ | $50.8_{\pm1.6}$ | $46.7_{\pm1.5}$ | $57.6_{\pm0.8}$ |
| LASER | $86.9_{\pm1.1}$ | $72.6_{\pm0.6}$ | $45.9_{\pm2.6}$ | $42.7_{\pm2.6}$ | $46.0_{\pm2.6}$ | $43.5_{\pm1.0}$ |
| GOKU | $86.8_{\pm0.3}$ | $73.6_{\pm0.2}$ | $\mathbf{72.4_{\pm2.2}}$ | $\mathbf{69.4_{\pm2.1}}$ | $\mathbf{68.8_{\pm1.4}}$ | $63.2_{\pm0.4}$ |
| FLAN | $\mathbf{88.3_{\pm0.9}}$ | $\mathbf{75.6_{\pm0.5}}$ | $55.6_{\pm3.0}$ | $51.9_{\pm3.1}$ | $54.5_{\pm2.9}$ | $\mathbf{65.1_{\pm0.6}}$ |

(b) Graph classification (Backbone: GCN on the left; GIN on the right)

| | **Backbone: GCN** | | | | **Backbone: GIN** | | | |
|---|---|---|---|---|---|---|---|---|
| | ENZYMES | IMDB | MUTAG | PROTEINS | ENZYMES | IMDB | MUTAG | PROTEINS |
| None | $27.1_{\pm1.6}$ | $49.5_{\pm1.0}$ | $70.3_{\pm2.1}$ | $71.4_{\pm1.0}$ | $33.5_{\pm1.3}$ | $67.7_{\pm1.4}$ | $76.1_{\pm3.1}$ | $69.5_{\pm1.4}$ |
| SDRF | $26.1_{\pm1.1}$ | $49.1_{\pm0.9}$ | $70.5_{\pm2.1}$ | $71.4_{\pm0.8}$ | $32.4_{\pm1.3}$ | $69.4_{\pm1.4}$ | $79.5_{\pm2.6}$ | $71.4_{\pm0.8}$ |
| FOSR | $27.4_{\pm1.1}$ | $49.6_{\pm0.8}$ | $75.6_{\pm1.7}$ | $72.3_{\pm0.9}$ | $28.8_{\pm1.0}$ | $70.6_{\pm1.3}$ | $74.8_{\pm1.5}$ | $73.7_{\pm0.8}$ |
| BORF | $24.7_{\pm1.0}$ | $50.1_{\pm0.9}$ | $75.8_{\pm1.9}$ | $71.0_{\pm0.8}$ | $31.4_{\pm1.5}$ | $70.5_{\pm1.3}$ | $78.2_{\pm1.6}$ | $71.9_{\pm1.3}$ |
| DR | – | $47.0_{\pm0.7}$ | $80.1_{\pm1.8}$ | $72.2_{\pm0.8}$ | – | $64.8_{\pm0.8}$ | $74.5_{\pm2.8}$ | $74.3_{\pm0.8}$ |
| GTR | $27.4_{\pm1.1}$ | $49.5_{\pm1.0}$ | $78.9_{\pm1.8}$ | $72.4_{\pm1.2}$ | $28.4_{\pm1.8}$ | $70.1_{\pm1.2}$ | $78.5_{\pm3.5}$ | $73.3_{\pm0.9}$ |
| LASER | $27.6_{\pm1.3}$ | $50.3_{\pm1.3}$ | $78.8_{\pm1.6}$ | $71.8_{\pm1.6}$ | $35.3_{\pm1.3}$ | $68.6_{\pm1.2}$ | $76.1_{\pm2.4}$ | $72.1_{\pm0.7}$ |
| GOKU | $27.6_{\pm1.2}$ | $49.8_{\pm0.7}$ | $81.0_{\pm2.0}$ | $71.9_{\pm0.8}$ | $33.8_{\pm1.2}$ | $71.3_{\pm0.9}$ | $78.4_{\pm2.5}$ | $73.9_{\pm1.0}$ |
| FLAN | $\mathbf{33.8_{\pm1.8}}$ | $\mathbf{54.8_{\pm1.6}}$ | $\mathbf{81.2_{\pm2.5}}$ | $\mathbf{74.3_{\pm1.7}}$ | $\mathbf{35.8_{\pm1.9}}$ | $\mathbf{72.0_{\pm1.3}}$ | $\mathbf{81.3_{\pm2.7}}$ | $\mathbf{74.2_{\pm1.7}}$ |

Table 2: Performance (%) on node and graph benchmarks. Best is in **bold**, second best underlined

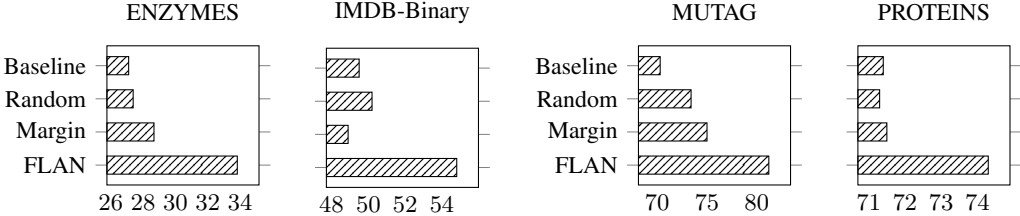

Figure 4: FLAN test accuracy vs. random shuffled $\hat{p}$ and a graph-level margin scalar, using a GCN backbone.

mechanism by quantifying both the intervention magnitude (e.g., $\|\mathbf{z} - \mathbf{h}^{(L)}\|$) and the resulting change in task margin in the graph classification task. Graphs with a higher long-range index $\rho_G$ receive stronger corrections from FLAN and achieve larger margin gains, showing that the layer adapts its intervention to the diagnostic's estimate of long-range demand and concentrates changes where they are most needed.

# 6 CONCLUSION

We reframed over-squashing as an task- and node-specific phenomenon. We (i) defined a margin-aligned sensitivity index for trained GNNs, (ii) showed it is predicted from topology via a sparse linear model, and (iii) found that margins co-vary with this sensitivity with dataset/backbone-dependent sign. Leveraging these findings, we introduced FLAN, a lightweight, rewiring-free readout layer that conditions on a structure-predicted proxy, improving accuracy without changing the graph. Our results open promising directions, including targeted rewiring at high-sensitivity nodes. In future work, we will study how this diagnostic can guide and complement graph rewiring methods.

## REPRODUCIBILITY STATEMENT

An anonymized code repository is linked at the end of the Introduction. All datasets, preprocessing steps, fixed splits, hyperparameters, and training/evaluation scripts are specified in the main text and in Appendix A.1, enabling full reproduction of our results.

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

## A APPENDIX

### A.1 EXPERIMENTAL SETUP FOR THE LONG-RANGE CAPTURE INDEX

We report here the GNN hyperparameters used to study the correlation between the long-range capture index $p_u$ and the task-aware classification margin across node and graph-level benchmarks in section 3. Our choices follow common evaluation protocols for rewiring methods with standard GNN backbones for both node classification (Pei et al., 2020; Attali et al., 2024a) and graph classification (Errica, 2023; Deac et al., 2022; Karhadkar et al., 2023; Wilson et al., 2024; Liang et al., 2025).

**Node classification.** We use two layers, dropout 0.5, learning rate 0.005, and early stopping with a patience of 100 epochs. Hidden dimensions are 32 for Texas, Wisconsin, and Cornell; 48 for Squirrel and Chameleon; and 16 for Cora and Citeseer. We adopt a fixed split with 60% of nodes for training, 20% for validation, and 20% for testing.

**Graph classification.** We use 4 layers, dropout 0.5, learning rate 0.001, hidden dimension 64, and early stopping with a patience of 100 epochs. Datasets are split into 80% training, 10% validation, and 10% testing.

For large graphs, we control the cost of computing the long-range index by using a random sample Jacobian estimation, as done in Bamberger et al. (2025), which makes the diagnostic scalable and stable in practice.

### A.2 ADDITIONAL ANALYSIS FOR CORRELATION BEHAVIOR

**Correlation with deeper GNN.** To assess the robustness of the correlation between the node margin $m_u$ and the long-range capture index $\rho_u$ with respect to model depth, we repeat the analysis using deeper 4-layer GCN, GAT, and GIN backbones. Across all datasets and architectures, the correlation values remain consistent with those obtained using shallower models, indicating that the phenomenon is stable under increased depth.

| Dataset | GCN: $\rho^{(1)}$ | GAT: $\rho^{(1)}$ | GIN: $\rho^{(1)}$ |
|---|---|---|---|
| Chameleon | $-0.3841 \pm 0.0526$ | $-0.3465 \pm 0.0633$ | $-0.1707 \pm 0.1272$ |
| Squirrel | $-0.3099 \pm 0.0644$ | $-0.2205 \pm 0.0800$ | $-0.2617 \pm 0.1006$ |
| Texas | $-0.5350 \pm 0.1086$ | $-0.3248 \pm 0.1353$ | $-0.3929 \pm 0.1465$ |
| Cornell | $-0.4430 \pm 0.1024$ | $-0.3680 \pm 0.1467$ | $-0.1322 \pm 0.1826$ |
| Wisconsin | $-0.4696 \pm 0.1147$ | $-0.3045 \pm 0.0842$ | $-0.0676 \pm 0.1424$ |
| Cora | $0.1910 \pm 0.0396$ | $0.1595 \pm 0.0431$ | $0.1372 \pm 0.0303$ |
| Citeseer | $-0.0063 \pm 0.0466$ | $0.0190 \pm 0.0316$ | $0.0176 \pm 0.0279$ |
| Amazon-Ratings | $-0.1799 \pm 0.0531$ | $-0.0850 \pm 0.0645$ | $-0.1408 \pm 0.0565$ |
| Roman-Empire | $-0.1811 \pm 0.0590$ | $-0.0342 \pm 0.0609$ | $-0.2034 \pm 0.0857$ |

Table 3: Correlation between the predictive margin $m_u$ and the long-range capture index $\rho_u$ for deeper (4-layer) GCN, GAT, and GIN backbones. The stability across architectures and datasets shows that the correlation behavior is robust to network depth.

**Limits of local signal for predicting $p_u$.** To further contextualize the results of the correlation in Table 1, we complement our analysis with two simple reference baselines that help clarify how much information about the model derived long-range index $p_u$ can be captured from local graph structure alone. Specifically, we evaluate (i) a degree-only predictor using the minimum, mean, and maximum degree of each node, and (ii) a random baseline obtained by shuffling the target values. As shown in Tables 4 and 5, structural measures commonly used in graph rewiring methods achieve consistently high test $R^2$ across datasets, indicating that they capture the topological factors most aligned with the long-range index $p_u$. In contrast, degree-only predictors perform poorly, and the random baseline yields strongly negative $R^2$, confirming that simple local degree statistics are insufficient to explain the model's long-range sensitivity.

| Dataset | Structural | Degree-only | Random |
|---|---|---|---|
| MUTAG | $0.9900 \pm 0.0057$ | $0.2028 \pm 0.1478$ | $-1.5234 \pm 0.3083$ |
| PROTEINS | $0.9395 \pm 0.0150$ | $0.1444 \pm 0.0884$ | $-1.0524 \pm 0.2995$ |
| ENZYMES | $0.8721 \pm 0.0411$ | $0.0754 \pm 0.0556$ | $-0.6743 \pm 0.3158$ |
| Cornell | $0.6745 \pm 0.0463$ | $-0.0060 \pm 0.0097$ | $-1.0397 \pm 0.1161$ |
| Texas | $0.5546 \pm 0.0474$ | $-0.7710 \pm 1.4327$ | $-0.9665 \pm 0.4239$ |
| Wisconsin | $0.4530 \pm 0.0644$ | $-0.0581 \pm 0.0664$ | $-1.1481 \pm 0.2667$ |
| Roman-Empire | $0.5196 \pm 0.0713$ | $-0.0101 \pm 0.0345$ | $-1.2238 \pm 0.4580$ |
| Amazon-Ratings | $0.7054 \pm 0.0511$ | $0.0875 \pm 0.0597$ | $-0.9520 \pm 0.1935$ |

Table 4: GCN — Test $R^2$ for structural predictors, degree-only baselines, and random baselines.

| Dataset | Structural | Degree-only | Random |
|---|---|---|---|
| MUTAG | $0.9835 \pm 0.0096$ | $0.2402 \pm 0.1531$ | $-1.5975 \pm 0.3761$ |
| PROTEINS | $0.8587 \pm 0.1441$ | $0.1300 \pm 0.0999$ | $-1.0031 \pm 0.3505$ |
| ENZYMES | $0.8659 \pm 0.0420$ | $0.0727 \pm 0.0553$ | $-0.6485 \pm 0.3047$ |
| Cornell | $0.7046 \pm 0.0629$ | $-0.0283 \pm 0.0701$ | $-0.7808 \pm 0.1697$ |
| Texas | $0.6786 \pm 0.0823$ | $0.0141 \pm 0.0283$ | $-1.2849 \pm 0.2970$ |
| Wisconsin | $0.6137 \pm 0.0339$ | $-0.0097 \pm 0.0224$ | $-0.7972 \pm 0.6848$ |
| Roman-Empire | $0.3947 \pm 0.0741$ | $-0.0016 \pm 0.0309$ | $-1.1918 \pm 0.3741$ |
| Amazon-Ratings | $0.6945 \pm 0.0505$ | $0.0747 \pm 0.0571$ | $-0.9480 \pm 0.1695$ |

Table 5: GIN — Test $R^2$ for structural predictors, degree-only baselines, and random baselines.

**Discussion on the choice of hop cutoffs.** While alternative choices for the boundary between short- and long-range interactions are possible, our results do not depend on selecting the distance-1 bin as the short-range component. In practice, redefining the cutoff for example using distances $(1, 2)$ or $(1, 2, 3)$ as the short-range part changes the absolute values of the index but leaves its correlation with the margin $m_u$ essentially unchanged. This stability arises because the relative ordering of nodes according to their distance-binned sensitivity distribution $\mathbf{S}_u^{(k)}$ is highly consistent across hop definitions. Thus, although our operational cutoff aligns with the locality of one-step message passing, the empirical relationship between long-range sensitivity and predictive margin is robust to how the hops are grouped.

For illustration, Table 6 reports the correlations obtained with a GCN backbone on several graph-classification datasets.

| Model & Dataset | Hop Range | Mean | Std |
|---|---|---|---|
| **GCN — MUTAG** | 1-hop only | $-0.4918$ | $0.0876$ |
| | 1–2 hops | $-0.4440$ | $0.0710$ |
| **GCN — PROTEINS** | 1-hop only | $-0.1809$ | $0.0550$ |
| | 1–2 hops | $-0.1676$ | $0.0420$ |
| **GCN — ENZYMES** | 1-hop only | $-0.2216$ | $0.1715$ |
| | 1–2 hops | $-0.1559$ | $0.1990$ |
| **GIN — MUTAG** | 1-hop only | $-0.4963$ | $0.0968$ |
| | 1–2 hops | $-0.4328$ | $0.1033$ |
| **GIN — PROTEINS** | 1-hop only | $-0.1479$ | $0.0649$ |
| | 1–2 hops | $-0.1134$ | $0.0579$ |
| **GIN — ENZYMES** | 1-hop only | $-0.2303$ | $0.1824$ |
| | 1–2 hops | $-0.0638$ | $0.1731$ |

Table 6: Correlation between predictive margin $m_u$ and long-range sensitivity index under different hop definitions.

Note that although the absolute magnitude of the long-range index varies across hop definitions (e.g., on MUTAG we obtain approximately $\rho_u^{(1)} \approx 0.17$, $\rho_u^{(1,2)} \approx 0.34$, and $\rho_u^{(1,2,3)} \approx 0.49$), the correlation with the margin remains stable.

**Dataset structural properties.** To make the structural differences across datasets explicit, we report in Table 7 (node-level benchmarks) and Table 8 (graph-level benchmarks) a detailed summary of key graph statistics, grouped by dataset family (homophilic vs. heterophilic) and by task type (node- vs. graph-level, including long-range benchmarks). These statistics help contextualize the long-range regime studied in our work, by linking each dataset's topology to the expected locality or non-locality of message passing.

| Dataset | #Nodes | #Edges | $\overline{\mathbf{d}}$ | $\mathrm{Std}(\mathbf{d})$ | $\max(\mathbf{d})$ | **Diam.** | **H** |
|---|---|---|---|---|---|---|---|
| Cora | 2.7k | 5.3k | 3.9 | 5.2 | 168 | 19 | 0.81 |
| Citeseer | 3.3k | 4.6k | 2.7 | 3.4 | 99 | 28 | 0.74 |
| Texas | 183 | 295 | 3.22 | 7.81 | 104 | 8 | 0.11 |
| Wisconsin | 251 | 466 | 3.71 | 7.95 | 122 | 8 | 0.20 |
| Cornell | 183 | 280 | 3.06 | 7.01 | 94 | 8 | 0.13 |
| Chameleon | 2.3k | 31.4k | 27.60 | 46.43 | 732 | 11 | 0.24 |
| Squirrel | 5.2k | 198k | 76.33 | 161.46 | 1905 | 10 | 0.22 |
| Roman Empire | 22.7k | 32.9k | 2.91 | 1.03 | 14 | 6824 | 0.05 |
| Amazon-Ratings | 24k | 93k | 7.60 | 6.00 | 132 | 46 | 0.38 |

Table 7: Node-level datasets: basic structural statistics. $\overline{d}$, $\mathrm{Std}(d)$, and $\max(d)$ denote mean, standard deviation, and maximum node degree; Diam. is the (graph) diameter; $H$ denotes homophily.

| Dataset | #Graphs | #Nodes ($\pm$) | #Edges ($\pm$) | $\overline{\mathbf{d}}$ | $\mathrm{Std}(\mathbf{d})$ | $\max(\mathbf{d})$ | **Diam.** |
|---|---|---|---|---|---|---|---|
| MUTAG | 188 | 17.9 ($\pm$4.6) | 19.8 ($\pm$5.7) | 2.2 | 0.7 | 3.0 | 8.2 |
| PROTEINS | 1113 | 39.1 ($\pm$45.8) | 72.8 ($\pm$84.6) | 3.7 | 0.9 | 5.8 | 11.6 |
| ENZYMES | 600 | 32.6 ($\pm$15.3) | 62.1 ($\pm$25.5) | 3.9 | 1.0 | 6.1 | 10.9 |
| REDDIT | 2000 | 429.6 ($\pm$554.1) | 497.8 ($\pm$623.0) | 2.3 | 8.9 | 217.4 | 9.7 |
| IMDB | 1000 | 19.8 ($\pm$10.1) | 96.5 ($\pm$105.6) | 8.9 | 2.8 | 18.8 | 1.9 |
| Peptides-func | 10873 | 151.5 ($\pm$84.1) | 154.3 ($\pm$86.0) | 2.0 | 0.8 | 3.0 | 57.1 |
| Peptides-struct | 10873 | 151.5 ($\pm$84.1) | 154.3 ($\pm$86.0) | 2.0 | 0.8 | 3.0 | 57.1 |

Table 8: Graph-level datasets: average structural statistics across graphs (mean $\pm$ standard deviation). $\overline{d}$, $\mathrm{Std}(d)$, and $\max(d)$ denote mean, standard deviation, and maximum node degree; Diam. denotes the average graph diameter.

# B ADDITIONAL EXPERIMENTS ON LONG-RANGE BENCHMARKS

We extend our analysis to the Long-Range Graph Benchmark (LRGB) (Dwivedi et al., 2022), including Peptides-struct, Peptides-func, and the synthetic Tree-Neighbors-Match dataset (Alon & Yahav, 2021). Because these datasets include regression and multi-label prediction tasks, the notion of margin must generalize beyond standard multi-class classification so that larger values consistently denote better predictions.

**Margin for Peptides-struct.** Since regression targets are standardized (zero mean, unit variance), we define:

$$m_G = -\log\left(\frac{1}{T}\sum_{t=1}^{T}|\hat{y}_t - y_t|\right), \qquad m_G \in (-\infty, 0].$$

where $T$ denotes the number of prediction targets

**Margin for Peptides-func (multi-label classification).** For logits $z_t$ and binary labels $y_t \in \{0, 1\}$:

$$m_G = \frac{1}{T}\sum_{t=1}^{T}(2y_t - 1)\, z_t.$$

The margin increases when the logits confidently align with the ground-truth labels, and decreases when the predictions contradict them.

Using these definitions, we compute the correlation between the long-range capture index $p_u$ and the graph-level margin $m_G$. As shown in Table 9, Peptides-struct and Peptides-func both exhibit positive correlations, indicating that better predictions are associated with a larger one-hop share. This suggests that, despite being part of a long-range benchmark, these tasks are effectively dominated by local interactions, consistent with prior work Bamberger et al. (2025).

| Dataset | GCN | GIN |
|---|---|---|
| Tree-Neighbors-Match | $-0.2964 \pm 0.0535$ | $-0.1806 \pm 0.0686$ |
| Peptides-struct | $0.3163 \pm 0.0395$ | $0.3343 \pm 0.0329$ |
| Peptides-func | $0.6963 \pm 0.0488$ | $0.5463 \pm 0.0612$ |

Table 9: Correlation between the long-range capture index $p_u$ and the margin $m_G$ on long-range benchmarks.

We also evaluate FLAN on Peptides-struct and Peptides-func using the expermental protocol of Nguyen et al. (2023); Wilson et al. (2024), and include two baselines: PANDA Choi et al. (2024), a rewiring-free method and a virtual-node augmentation. Despite the local nature of the tasks, FLAN improves performance across both datasets. Since graphs with lower long-range sensitivity achieve higher margins, adjusting embeddings according to their predicted structural index $p_G$ reinforces the region of representation space most aligned with task-specific signals.

| Model | Peptides-func (AP $\uparrow$) | Peptides-struct (MAE $\downarrow$) |
|---|---|---|
| GCN | $0.5029 \pm 0.0058$ | $0.3587 \pm 0.0006$ |
| + SDRF | $0.5041 \pm 0.0026$ | $0.3559 \pm 0.0010$ |
| + FoSR | $0.4534 \pm 0.0090$ | $0.3003 \pm 0.0007$ |
| + EGP | $0.4972 \pm 0.0023$ | $0.3001 \pm 0.0013$ |
| + CGP | $0.5106 \pm 0.0014$ | $0.2931 \pm 0.0006$ |
| + VN | $0.5022 \pm 0.0014$ | $0.3241 \pm 0.0016$ |
| + PANDA | $0.5188 \pm 0.0022$ | $0.3098 \pm 0.0011$ |
| **+ FLAN** | $\mathbf{0.5479 \pm 0.0041}$ | $\mathbf{0.2724 \pm 0.0019}$ |
| GIN | $0.5124 \pm 0.0055$ | $0.3544 \pm 0.0014$ |
| + SDRF | $0.5122 \pm 0.0061$ | $0.3515 \pm 0.0011$ |
| + FoSR | $0.4584 \pm 0.0079$ | $0.3008 \pm 0.0014$ |
| + EGP | $0.4926 \pm 0.007$ | $0.3034 \pm 0.0027$ |
| + VN | $0.5137 \pm 0.0060$ | $0.3197 \pm 0.0021$ |
| + CGP | $0.5159 \pm 0.0059$ | $0.2910 \pm 0.0011$ |
| + PANDA | $0.5214 \pm 0.0068$ | $0.3003 \pm 0.0019$ |
| **+ FLAN** | $\mathbf{0.5375 \pm 0.0041}$ | $\mathbf{0.2886 \pm 0.0021}$ |

Table 10: FLAN compared to rewiring and virtual-node baselines on Peptides-func and Peptides-struct.

**Additional LRGB evaluation details** Following the tuning recommendations of Tönshoff et al. (2024), we additionally evaluate our approach under a more exhaustive hyperparameter search for the GCN backbone. Concretely, we vary (i) the use of Batch Normalization (`on`/`off`) and (ii) the prediction head, replacing the linear classifier with an MLP of depth $d \in \{1, 2, 3\}$, while using a base learning rate $\mathrm{lr} = 10^{-3}$ together with a learning-rate schedule. The corresponding results are reported in Table 11, and show that FLAN consistently improves the tuned GCN baseline.

Note that to control the computational cost of our long-range index on large graphs, we use a random-sample Jacobian estimator, following Bamberger et al. (2025). This approximation makes the diagnostic scalable and stable in practice.

Beyond accuracy, our method remains computationally lightweight,depending on the dataset and the considered Graph Transformer variants, we obtain near state-of-the-art performance while reducing the overall runtime by approximately $45\%$ to $95\%$.

Table 11: LRGB results on Peptides-func (AP ↑) and Peptides-struct (MAE ↓). **Ours** are mean ± std over 4 runs; other methods are reported numbers from Tönshoff et al. (2024). Best baseline per dataset is in **bold**.

| Group | Method | Peptides-func AP ↑ | Peptides-struct MAE ↓ |
|---|---|---|---|
| MPNNs | GCN (ours) | $0.6655 \pm 0.0039$ | $0.2558 \pm 0.0024$ |
| Multi-hop GNNs | DIGL+MPNN+LapPE | $0.6830 \pm 0.0026$ | $0.2616 \pm 0.0018$ |
| | MixHop-GCN+LapPE | $0.6843 \pm 0.0049$ | $0.2614 \pm 0.0023$ |
| | DRew-GCN+LapPE | $\mathbf{0.7150 \pm 0.0044}$ | $0.2536 \pm 0.0015$ |
| Graph Transformers | Transformer+LapPE | $0.6326 \pm 0.0126$ | $0.2529 \pm 0.0016$ |
| | SAN+LapPE | $0.6384 \pm 0.0121$ | $0.2683 \pm 0.0043$ |
| | GraphGPS+LapPE | $0.6535 \pm 0.0041$ | $0.2500 \pm 0.0005$ |
| | GPS | $0.6534 \pm 0.0091$ | $0.2509 \pm 0.0014$ |
| | CRAWL | $0.7074 \pm 0.0032$ | $0.2506 \pm 0.0022$ |
| | GRIT | $0.6988 \pm 0.0082$ | $0.2460 \pm 0.0012$ |
| | Graph ViT | $0.6942 \pm 0.0075$ | $\mathbf{0.2449 \pm 0.0016}$ |
| | G-MLPMixer | $0.6921 \pm 0.0054$ | $0.2475 \pm 0.0015$ |
| Ours | **GCN + FLAN** | $0.6868 \pm 0.0040$ | $\mathbf{0.2450 \pm 0.0026}$ |

## C  FLAN'S EFFECT ON LONG-RANGE SENSITIVITY

In this section, we visualize how graph embeddings change under FLAN as a function of the graph-level long-range index $\rho_G$. Across datasets (Figures 10, 6, and 7), we observe a clear trend: graphs with higher $\rho_G$ undergo stronger embedding shifts and achieve larger margin gains. This pattern indicates that FLAN does not apply a uniform correction but adapts the intensity of its intervention to the diagnostic's estimate of long-range demand. Importantly, the largest modifications occur precisely for graphs where local evidence dominates and long-range contributions are under-represented, confirming that the diagnostic successfully identifies the regimes where intervention is most beneficial.

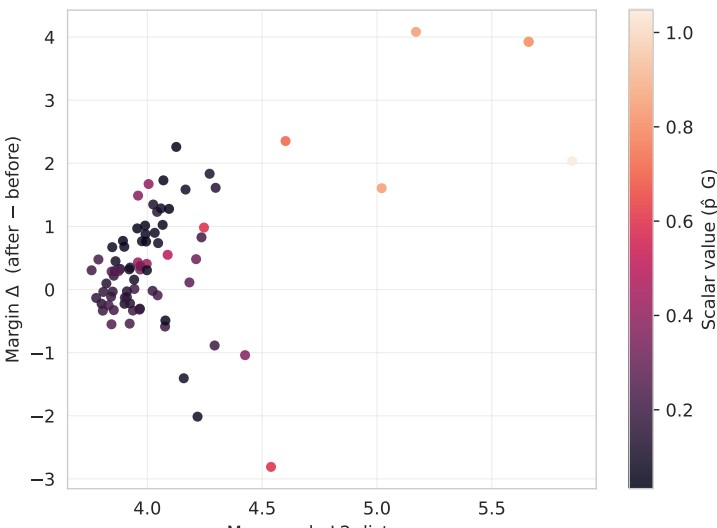

Figure 5: Analysis of FLAN adjustments on PROTEINS: the Euclidean distance $\|\mathbf{z} - \mathbf{h}^{(L)}\|$ (difference between the embedding with FLAN, $\mathbf{z}$, and the backbone embedding without FLAN, $\mathbf{h}^{(L)}$) and the margin gain are plotted against the long-range index $\rho_G$, showing larger adjustments and stronger improvements as $\rho_G$ increases.

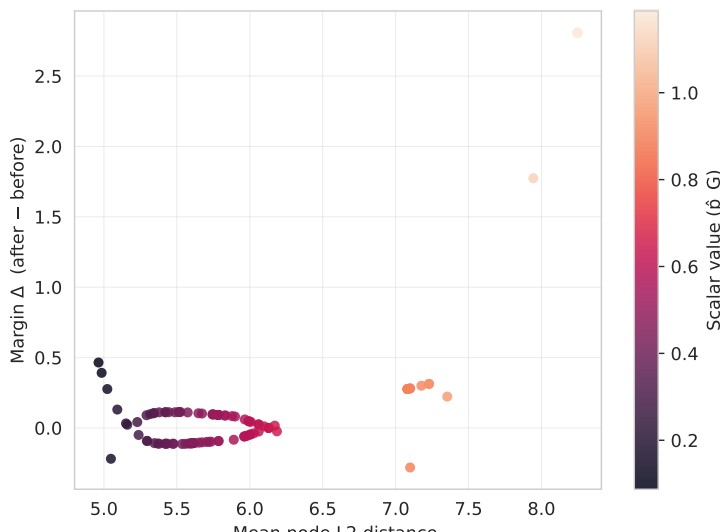

Figure 6: Analysis of FLAN adjustments on IMDB-Binary: the Euclidean distance $\|\mathbf{z} - \mathbf{h}^{(L)}\|$ (difference between the embedding with FLAN, $\mathbf{z}$, and the backbone embedding without FLAN, $\mathbf{h}^{(L)}$) and the margin gain are plotted against the long-range index $\rho_G$, showing larger adjustments and stronger improvements as $\rho_G$ increases.

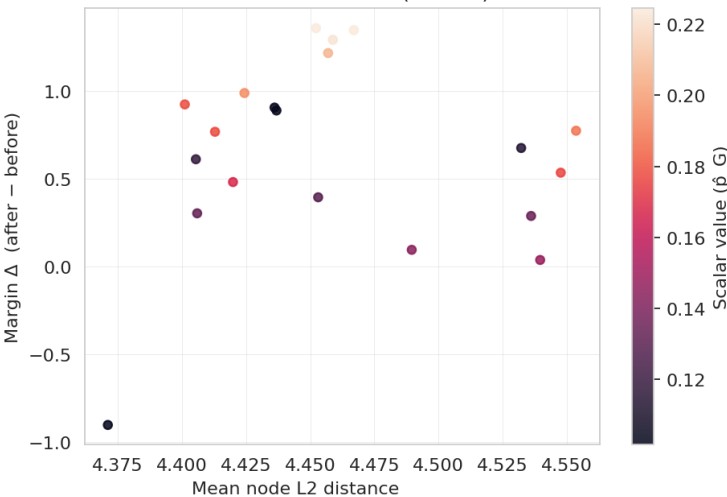

Figure 7: Analysis of FLAN adjustments on MUTAG: the Euclidean distance $\|\mathbf{z} - \mathbf{h}^{(L)}\|$ (difference between the embedding with FLAN, $\mathbf{z}$, and the backbone embedding without FLAN, $\mathbf{h}^{(L)}$) and the margin gain are plotted against the long-range index $\rho_G$, showing larger adjustments and stronger improvements as $\rho_G$ increases.

To show that FLAN focuses on the graphs that are most sensitive to long-range degradation, we examine how the margin improvement varies across different values of the long-range index $p_G$. As shown in Table 12, the performance gains are concentrated on the graphs with the highest $p_G$, i.e., those whose structure is most affected by long-range dependencies. This indicates that our method specifically benefits the graphs for which message passing is most challenged by long-range effects. The table quantifies this behavior by reporting how the margin improvement $\Delta m$ increases when restricting evaluation to the top 50%, 25%, and 10% most long-range-sensitive graphs.

| Dataset | Backbone | $\Delta m$ (all) | $\Delta m$ (Top50%) | $\Delta m$ (Top25%) | $\Delta m$ (Top10%) |
|---|---|---|---|---|---|
| MUTAG | GCN | +0.3944 | +0.1027 | +0.1101 | +0.0921 |
| | GIN | +0.1617 | +0.0428 | +0.1091 | +0.1547 |
| ENZYMES | GCN | +0.2675 | +0.2424 | +0.4382 | +0.9524 |
| | GIN | +0.0526 | -0.0081 | +0.3468 | +0.4252 |
| PROTEINS | GCN | +0.0842 | +0.1217 | +0.4729 | +1.1336 |
| | GIN | +0.1917 | +0.1790 | +0.4149 | +0.8718 |
| IMDB | GCN | +0.0180 | +0.0580 | +0.0854 | +0.1258 |
| | GIN | +0.4964 | +0.8223 | +1.5778 | +2.4207 |

Table 12: Margin improvement $\Delta m$ across the full dataset and on subsets of graphs with the highest long-range index $p_G$.

To further validate that the gains come from conditioning on long-range sensitivity rather than from the estimation procedure, we include an ablation where FLAN is given access to the *true* backbone-derived $p_G$ (denoted FLAN⋆). As shown in Table 13, the improvements obtained with FLAN⋆ are nearly identical to those obtained with the predicted index, confirming that the diagnostic quantity is responsible for the performance gains.

| Backbone | Method | ENZYMES | IMDB | MUTAG | PROTEINS |
|---|---|---|---|---|---|
| GCN | FLAN | $33.8 \pm 1.8$ | $54.8 \pm 1.6$ | $81.2 \pm 2.5$ | $74.3 \pm 1.7$ |
| | FLAN⋆ | $34.7 \pm 1.7$ | $55.1 \pm 1.5$ | $81.6 \pm 1.8$ | $74.5 \pm 2.1$ |
| GIN | FLAN | $35.8 \pm 1.9$ | $72.0 \pm 1.3$ | $81.3 \pm 2.7$ | $74.2 \pm 1.7$ |
| | FLAN⋆ | $36.3 \pm 1.5$ | $72.9 \pm 1.8$ | $81.1 \pm 2.0$ | $74.5 \pm 2.3$ |

Table 13: Comparison between FLAN and FLAN⋆, where the latter uses the true long-range index $p_G$ computed from Jacobian sensitivities.

**Sensitivity Capture and FLAN-Accuracy Gains** In this section, we present (i) the cumulative sensitivity capture profile $\rho_u^{(1,\dots,k)}$ as a function of the hop radius $k$ (blue, left axis), averaged over graphs with a $\pm 1$ standard-deviation, and (ii) the corresponding test-accuracy gain $\Delta(k)$ of FLAN over the baseline (orange, right axis), reported in percentage points. Figures **??** report results averaged over 10 random runs. When $\rho_u^{(1,\dots,k)}$ saturates quickly (i.e., approaches 1 for small $k$), it indicates that the margin-aligned sensitivity mass is predominantly concentrated at short range (typically IMDB-Binary). Conversely, slow saturation implies that a substantial fraction of this sensitivity is distributed over more distant hops, reflecting stronger long-range dependencies (e.g., PROTEINS or ENZYMES). The curve $\Delta(k)$ indicates the scale $k$ at which the signal $\hat{p}_G$ is most informative for FLAN: a peak at small $k$ suggests that the most discriminative information lies at local to mid-range neighborhoods, whereas a decline at larger $k$ is typically consistent with saturation of $\rho^{(1,\dots,k)}$.

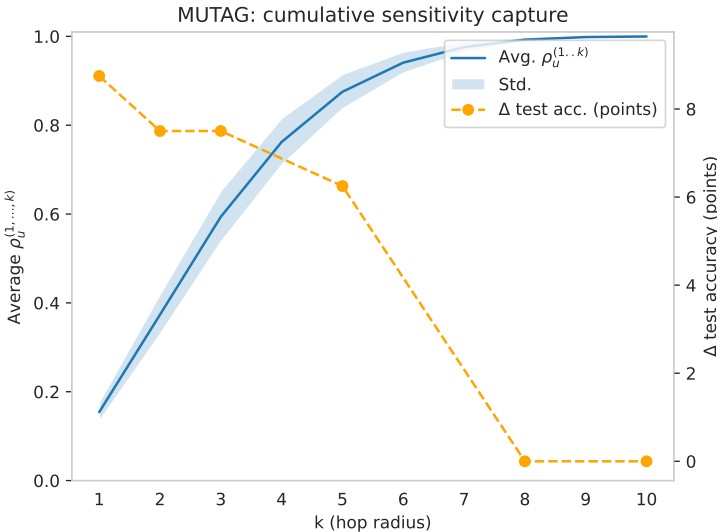

Figure 8: Cumulative sensitivity capture $\rho_u^{(1,...,k)}$ (blue, mean±std) and FLAN gain $\Delta(k)$ in points (orange) vs. hop radius. on MUTAG dataset.

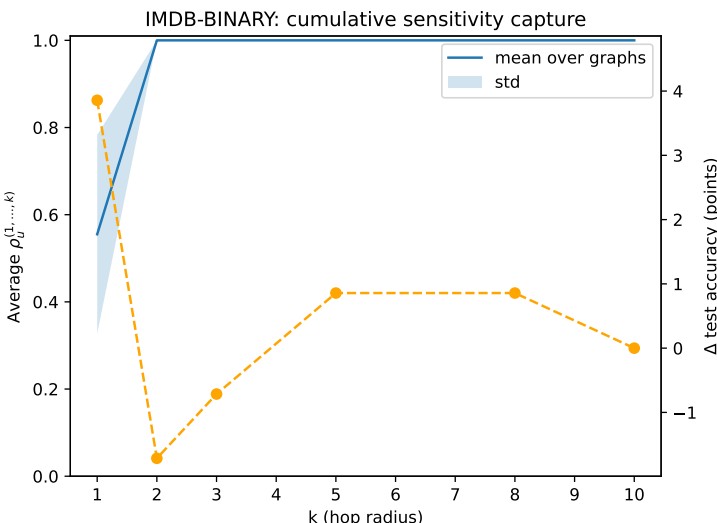

Figure 9: Cumulative sensitivity capture $\rho_u^{(1,...,k)}$ (blue, mean±std) and FLAN gain $\Delta(k)$ in points (orange) vs. hop radius. on IMDB dataset.

## D   TIME COMPARISON

Table 14 reports the average preprocessing time per graph; for FLAN we report the end-to-end cost of producing the conditioning scalar $\hat{p}_G$ (Jacobian–margin evaluation + structural indicators + Lasso fit), to match the per-graph preprocessing measured for rewiring baselines.

FLAN's preprocessing cost stays in the millisecond range per graph and is comparable to graph expanders such as EGP/CGP, while being $10^1$–$10^3\times$ faster than heavier rewiring methods (e.g., BORF, SDRF/FoSR, and GTR).

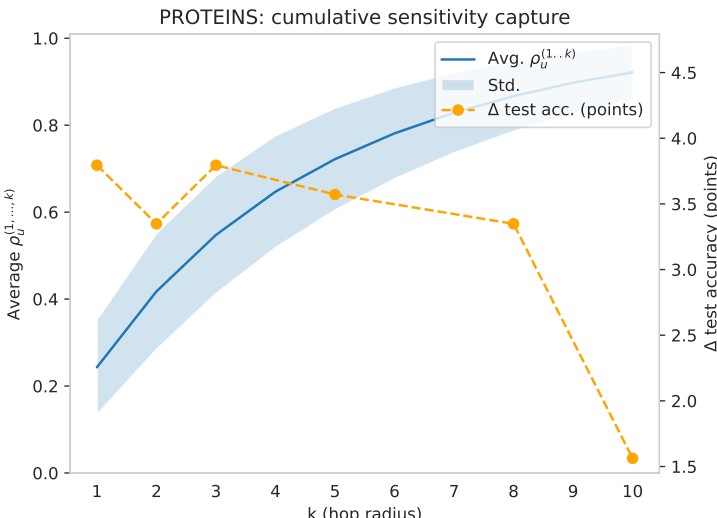

Figure 10: Cumulative sensitivity capture $\rho_u^{(1,\ldots,k)}$ (blue, mean±std) and FLAN gain $\Delta(k)$ in points (orange) vs. hop radius. on PROTEINS dataset.

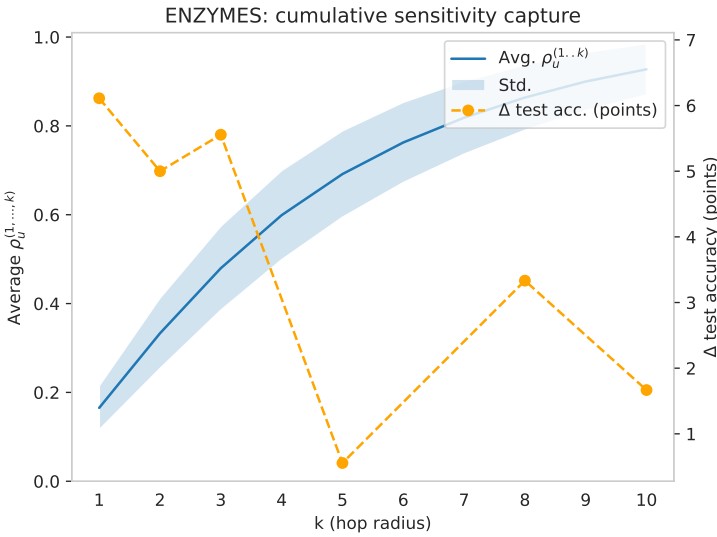

Figure 11: Cumulative sensitivity capture $\rho_u^{(1,\ldots,k)}$ (blue, mean±std) and FLAN gain $\Delta(k)$ in points (orange) vs. hop radius. on ENZYMES dataset.

| Model | IMDB-Binary | MUTAG | ENZYMES | PROTEINS |
|-------|-------------|-------|---------|----------|
| SDRF | 5.13257 | 0.669701 | 1.71482 | 3.02873 |
| FoSR | 4.54634 | 4.71567 | 4.56855 | 5.04358 |
| BORF | 465.408 | 53.7069 | 179.573 | 351.173 |
| GTR | 3.39839 | 1.54127 | 2.87399 | 6.49714 |
| PANDA | 0.789759 | 0.246243 | 0.278594 | 0.248043 |
| EGP | 0.0185697 | 0.00446963 | 0.0163198 | 0.0393348 |
| CGP | 0.0211341 | 0.00438905 | 0.0166841 | 0.0348585 |
| FLAN | 0.017668 | 0.013909 | 0.016429 | 0.027119 |

Table 14: Comparison of the preprocessing time to construct each graph rewiring method compared to our FLAN method (in seconds per graph). Table taken from Wilson et al. (2024).

