# OpenReview forum: "When Do Distant Dependencies Matter? Diagnostics for Long-Range Propagation in GNNs"
_ICLR.cc/2026/Conference — Submitted to ICLR 2026_

### Official Review · Reviewer_REWK · 2025-10-27

**Soundness:** 2
**Presentation:** 2
**Contribution:** 2
**Rating:** 4
**Confidence:** 3

**Summary:**

- This paper introduces a measure of range for trained GNNs: the proportion of accurate classification confidence ('margin') sensitivity (Jacobian) that is confined to the one-hop neighbourhood; for node- and graph-classification tasks.
- The authors show that an estimate of this 'long-range capture index' can be obtained from a lasso regression over several 'structural indicators' based purely on graph topology and commonly used in graph rewiring methods — and that there is a correlation between their LR capture index and this structure-only estimate
- The authors introduce an alternative graph readout, FLAN, that re-weights and shifts output layer node features by their node-level estimated (i.e. structure-only) LR indices, without requiring rewiring, architecture changes or additional (significant) compute/time
- Experiments on several classification datasets show a minimal benefit for node classification tasks (for which the estimated index correlates weakly) and a modest but consistent benefit for graph classification tasks (for which the estimated index correlates more strongly)

**Strengths:**

- [S1] The paper is reasonably well-written
- [S2] The long-range capture index is intuitive
- [S3] There are existing long-range sensitivity measures using the Jacobian, but baking in model performance with classification margin is a neat idea
- [S4] The FLAN layer provides a modest performance boost on TUD graph classification tasks with minimal compute requirements compared to rewiring methods

**Weaknesses:**

- [W1] A core contribution of this paper is the introduction of a 'diagnostic of long-range sensitivity'. A few points on this:
	- [W1.1] The long-range capture index is a "one-hop dominance score" for a given node, and the authors argue that a low index indicates that there are more long-range interactions. However, there is no distinction between a node being dominated by, e.g., nodes at the 2-hop distance and a node being dominated by nodes at greater distances, or a variety of distances. I would argue that a 2-hop dominated node is no more 'long-range' than one dominated by the 1-hop
	- [W1.2] What is this index actually for? Is it intended as a measure of how 1-hop-dominated a model is, or how 1-hop-dominated a task is? It seems to me that if we find that a model is highly 1-hop dominated, we don't know if that is because the model has failed to capture long-range interactions or if it is because the task itself is 1-hop dominated
	- [W1.3] A Jacobian-based long-range measure is not novel; I would like to see a discussion of [1, 4]
	- [W 1.4] Later in the paper it is argued that a structural proxy index correlates strongly enough with the long-range index to be used as a substitute. Does this not somewhat undermine the long-range index as a contribution?

- [W2] I am surprised that a paper that purports to be primarily about long-range interactions does not make use of datasets explicitly described as long-range, or mention graph Transformers, or multi-hop MPNNs, or any other architectures that explicitly address long-range interactions, other than rewiring methods
	- [W2.1] The paper seems focused only on rewiring methods and over-squashing; indeed, it seems to be more a paper about over-squashing in general than it is about long-range interactions
	- [W2.2] Though it is a flawed benchmark [2], experiments on the Long Range Graph Benchmark [3], or at minimum, a discussion, would seem essential for a LR-focused paper
	- [W2.3] Other long-range datasets have been introduced recently [4, 5] that should also be discussed
	- [W2.4] Line 397: you claim that the graph classification benchmarks of TUD 'require the propagation of LR dependencies', and indeed this is a quote from Karhadkar et al. 2023, but that paper only cites the original Morris et al. 2020, and AFAIK that paper does not make this claim of long-rangedness. Could you justify what makes these tasks long-range? IMDB, for example, appears to have an average diameter of <2 hops.  "Their structures are tightly coupled to the downstream task" — what do you mean by this? Is this not the case for all graph tasks?

- [W3] Given that much of the paper, including FLAN, is built on the idea that the structural proxy index correlates strongly with the long-range capture index, I was a little underwhelmed by Table 1 — many of the datasets correlate quite weakly, and many are not robust over backbones as you claim.

- [W4] I would have liked some more details in the Appendix about the datasets used here — their characteristics (size, diameter, average node pair distance, etc). I would suggest including a table in the Appendix showing such characteristics. In tables and figures I would also group related datasets under a header explaining their type/significance, e.g. heterophilic, small homophilic, 'long-range', graph- and node-level, etc
- [W5] I attempted to run the code and it worked for the MUTAG dataset only. I did not spend much time on this but it appears non-trivial to reproduce all experiments for other datasets. Even for MUTAG the results from the code provided do not reproduce those in the paper exactly. There is no README or documentation.
- [W6] Line 246: "On heterophilic graphs, GCN and GAT exhibit consistently negative correlations". RomanEmpire and AmazonRatings are both heterophilic and the correlation in Figure 2 is negligible
- [W7] You make no mention of over-smoothing; 'computational bottlenecks' described at line 143 would appear to correspond to over-smoothing, but there is no discussion of it or of over-smoothing literature
---
[1] Bamberger, Jacob, et al. "On Measuring Long-Range Interactions in Graph Neural Networks." arXiv preprint arXiv:2506.05971 (2025).

[2] Tönshoff, Jan, et al. "Where did the gap go? reassessing the long-range graph benchmark." arXiv preprint arXiv:2309.00367 (2023).

[3] Dwivedi, Vijay Prakash, et al. "Long range graph benchmark." Advances in Neural Information Processing Systems 35 (2022): 22326-22340.

[4] Liang, Huidong, et al. "Towards Quantifying Long-Range Interactions in Graph Machine Learning: a Large Graph Dataset and a Measurement." arXiv preprint arXiv:2503.09008 (2025).

[5] Zhou, Dongzhuoran, Evgeny Kharlamov, and Egor V. Kostylev. "GLoRa: A Benchmark to Evaluate the Ability to Learn Long-Range Dependencies in Graphs." The Thirteenth International Conference on Learning Representations. 2025.

**Questions:**

- [Q1] Line 387: "The sensitivity index compresses long-range demand into a single axis that is highly predictive of where the baseline fails. Because the dominant error varies monotonically with $p_u$, this rank-1 translation plus diagonal reweighting is a minimal intervention that corrects the under-performing p regime."
	- I don't understand what this means. Could you elaborate?
- [Q2] In Figure 1, is each scatter point a different node? What is $\bar{\rho}_u$? This appears to be the only use of bar notation.
	- Notation is a bit inconsistent; there are several places where $\rho$ is used for the LR capture index, but Section 3 leads me to believe that the index should be represented by either $p_u$ for a single node, or $\rho_G$ for a graph (on this point, why mix $p$ and $\rho$ at all?)
	- You also use $S$ for both the node-level structural indicator in (7) and for source/aggregated sensitivity in (3)
- [Q3] It is an interesting result that the structural proxy index recovers $p_u$ so closely for some datasets, e.g. MUTAG. Can you elaborate on why you think this might be?

---

> ### Author Response · Authors · 2025-11-19
> **Rebuttal part 1**
>
> We thank the reviewer for their careful evaluation and constructive feedback, which helped us strengthen both the clarity and the scope of the paper.
>
> **(W1.1) Definition of short vs. long-range interactions.**
>
> While alternative choices for the boundary between short and long-range interactions are possible, our results do not depend on selecting the distance-$1$ bin as the short-range component. In practice, redefining the cutoff for example using distance $(1,2)$ or distances $(1,2,3)$ as the short-range part changes the absolute values of the index but leaves its correlation with the margin $m_u$ essentially unchanged.
> This stability arises because the relative ordering of nodes according to their distance-binned sensitivity distribution $\{S_u(k)\}$ is highly consistent across hop definitions. Thus, although our operational cutoff aligns with the locality of one-step message passing, the empirical relationship between long-range sensitivity and predictive margin is robust to how the hops are grouped.
>
> For illustration, the table below reports the correlations obtained with a GCN backbone on graph classification datasets:
>
> | **Model & Dataset** | **Hop Range**       | **Mean** | **Std** |
> |---------------------|----------------------|----------|---------|
> | **GCN — MUTAG**     | 1-hop only           | -0.4918  | 0.0876  |
> |                     | 1–2 hops             | -0.4440  | 0.0710  |
> | **GCN — PROTEINS**  | 1-hop only           | -0.1809  | 0.0550  |
> |                     | 1–2 hops             | -0.1676  | 0.0420  |
> | **GCN — ENZYMES**   | 1-hop only           | -0.2216  | 0.1715  |
> |                     | 1–2 hops             | -0.1559  | 0.1990  |
> | **GIN — MUTAG**     | 1-hop only           | -0.4963  | 0.0968  |
> |                     | 1–2 hops             | -0.4328  | 0.1033  |
> | **GIN — PROTEINS**  | 1-hop only           | -0.1479  | 0.0649  |
> |                     | 1–2 hops             | -0.1134  | 0.0579  |
> | **GIN — ENZYMES**   | 1-hop only           | -0.2303  | 0.1824  |
> |                     | 1–2 hops             | -0.0638  | 0.1731  |
>
> Note that even though the absolute values of the long-range index vary across hop definitions (e.g., on MUTAG the average values are approximately $\rho_u^{(1)} \approx 0.17$, $\rho_u^{(1,2)} \approx 0.34$, and $\rho_u^{(1,2,3)} \approx 0.49$), the correlation with the margin remains stable.
>
> ---
>
> **(W1.2) Clarification of the Long-Range Capture Index (long-range vs. task locality).**
>
> If the model fails to capture long-range information, the correlation between the index and the margin becomes negative (as observed in heterophilic graphs). Conversely, when the task is inherently local, the correlation tends to be close to zero or even positive (as in homophilic datasets). In general, the index reveals whether performance depends primarily on long-range interactions or is dominated by short-range locality, and it can be used to study both dataset-specific locality patterns and the behaviour of different GNN architectures.
>
> ---
>
>
> **(W1.3)  Clarification of the Long-Range Capture Index (novelity of our measure).**
>
> Our diagnostic index measures a task-aligned, margin-aware form of long-range sensitivity, whereas the range measure in [1] is operator-aligned and intentionally task-agnostic. Their metric quantifies how far Jacobian/Hessian influence can propagate across the graph, but it does not indicate whether such influence is relevant to the decision boundary. In contrast, our index isolates the share of margin-relevant sensitivity captured at one hop versus longer ranges, which allows us to determine when long-range propagation helps or hurts performance. We will clarify this distinction in the revised version, including a brief discussion comparing the intent, scope, and interpretability of both metrics in the context of long-range behavior.
>
>  [1] Bamberger, J., Gutteridge, B., le Roux, S., Bronstein, M. M., & Dong, X. (2025). On Measuring Long-Range Interactions in Graph Neural Networks. In Proceedings of the 42nd International Conference on Machine Learning

---

> > ### Author Response · Authors · 2025-11-19
> > **Rebuttal part 2**
> >
> > **(W1.4)  On whether the structural proxy undermines the contribution of the long-range index.**
> >
> > Thank you for raising this point. The existence of a structural proxy does not weaken the contribution of our long-range index for two main reasons.
> >
> > First, our goal is precisely to understand whether aspects of long-range sensitivity defined from the model’s Jacobian can be predicted from the underlying graph structure. The strong correlations we report show that some structural descriptors commonly used in rewiring (PageRank, Forman curvature, commute-time resistance) contain enough information to approximate the index. This validates the idea that long-range behaviour in GNNs is tightly linked to specific structural properties of the graph.
> >
> > Second, the proxy does not make the index redundant. It only succeeds because the index first provides a principled, task-aligned quantity to regress against. Moreover, the proxy does not arise from arbitrary structural features : simple statistics such as min/mean/max degree or random baseline perform extremely poorly, which confirms that the index captures information that is not accessible through naive graph descriptors (See Table below). Only more informative structural quantities correlate well.
> >
> > | **Dataset**  | **Structural**     | **Degree-only**        | **Random**        |
> > |--------------------|---------------------------|---------------------------|---------------------------|
> > | MUTAG              | 0.9900 ± 0.0057           | 0.2028 ± 0.1478           | −1.5234 ± 0.3083          |
> > | PROTEINS           | 0.9395 ± 0.0150           | 0.1444 ± 0.0884           | −1.0524 ± 0.2995          |
> > | ENZYMES            | 0.8721 ± 0.0411           | 0.0754 ± 0.0556           | −0.6743 ± 0.3158          |
> > | Cornell            | 0.6745 ± 0.0463           | −0.0060 ± 0.0097          | −1.0397 ± 0.1161          |
> > | Texas              | 0.5546 ± 0.0474           | −0.7710 ± 1.4327          | −0.9665 ± 0.4239          |
> > | Wisconsin          | 0.4530 ± 0.0644           | −0.0581 ± 0.0664          | −1.1481 ± 0.2667          |
> > | Roman-Empire       | 0.5196 ± 0.0713           | −0.0101 ± 0.0345          | −1.2238 ± 0.4580          |
> > | Amazon-Ratings     | 0.7054 ± 0.0511           | 0.0875 ± 0.0597           | −0.9520 ± 0.1935          |
> >
> > A similar trend is observed when using a GIN backbone.
> > We will clarify this point in the updated version.

---

> > > ### Author Response · Authors · 2025-11-19
> > > **Rebuttal part 3**
> > >
> > > **(W2) Additional experiments on long-range benchmarks.**
> > >
> > > We extended our study to the Long-Range Graph Benchmark (LRGB) [2], including the Peptides-struct and Peptides-func datasets, as well as the synthetic Tree-Neighbors-Match dataset [3]. For the LRGB datasets, the notion of “margin’’ must generalize beyond standard multi-class classification. We therefore redefine it so that a larger value always indicates better predictions, regardless of the task type. These datasets include both multi-label graph classification (Peptides-func) and graph-level regression (Peptides-struct).
> > >
> > > Since regression targets in Peptides-struct are standardized (zero mean, unit variance), we define the graph-level margin as:
> > >
> > > $$
> > > m_G = -\log\left( \frac{1}{T}\sum_{t=1}^T |\hat{y}_t - y_t| \right), \quad m_G \in (-\infty, 0].
> > > $$
> > >
> > > The margin approaches $0$ when the prediction error approaches $0$, and decreases toward $-\infty$ otherwise.
> > >
> > > For Peptides-func, we define the margin as the average signed logit:
> > >
> > > $$
> > > m_G = \frac{1}{T} \sum_{t=1}^T (2y_t - 1)\, z_t, \quad m_G \in (-\infty, +\infty).
> > > $$
> > >
> > > This quantity increases when logits $z_t$ confidently align with ground-truth labels, and decreases when predictions contradict them.
> > >
> > > We report the correlation between the margin $m_G$ and the long-range capture index $p_u$ on the long-range benchmarks. Both Peptides-struct and Peptides-func exhibit a positive correlation, indicating that models perform better when they rely less on distant information. This suggests that, despite being part of a long-range benchmark, these tasks are effectively dominated by local interactions, consistent with prior work [1].
> > >
> > > | **Dataset**           | **GCN**               | **GIN**                |
> > > |-----------------------|-----------------------|------------------------|
> > > | Tree-Neighbors-Match | −0.2964 ± 0.0535      | −0.1806 ± 0.0686       |
> > > | Peptides-struct      | 0.3163 ± 0.0395       | 0.3343 ± 0.0329        |
> > > | Peptides-func        | 0.6963 ± 0.0488       | 0.5463 ± 0.0612        |
> > >
> > > We also evaluate the proposed FLAN model on Peptides-struct and Peptides-func, following the experimental protocol of [4] and [5]. We include two baselines: PANDA [6] and a virtual-node augmentation. Despite the local nature of these tasks, FLAN still improves performance: since graphs with lower long-range sensitivity achieve higher margins, adjusting embeddings according to their predicted structural index $p_G$ naturally reinforces the part of the representation space best aligned with these tasks.
> > >
> > > | **Model**        | **Peptides-func (AP ↑)** | **Peptides-struct (MAE ↓)** |
> > > |------------------|---------------------------|------------------------------|
> > > | GCN              | 0.5029 ± 0.0058           | 0.3587 ± 0.0006              |
> > > | + SDRF           | 0.5041 ± 0.0026           | 0.3559 ± 0.0010              |
> > > | + FoSR           | 0.4534 ± 0.0090           | 0.3003 ± 0.0007              |
> > > | + EGP            | 0.4972 ± 0.0023           | 0.3001 ± 0.0013              |
> > > | + CGP            | 0.5106 ± 0.0014           | 0.2931 ± 0.0006              |
> > > | + VN             | 0.5022 ± 0.0014           | 0.3241 ± 0.0016              |
> > > | + PANDA          | 0.5188 ± 0.0022           | 0.3098 ± 0.0011              |
> > > | **+ FLAN**       | **0.5479 ± 0.0041**       | **0.2724 ± 0.0019**          |
> > > |                  |                           |                              |
> > > | GIN              | 0.5124 ± 0.0055           | 0.3544 ± 0.0014              |
> > > | + SDRF           | 0.5122 ± 0.0061           | 0.3515 ± 0.0011              |
> > > | + FoSR           | 0.4584 ± 0.0079           | 0.3008 ± 0.0014              |
> > > | + EGP            | 0.4926 ± 0.0070           | 0.3034 ± 0.0027              |
> > > | + VN             | 0.5137 ± 0.0060           | 0.3197 ± 0.0021              |
> > > | + CGP            | 0.5159 ± 0.0059           | 0.2910 ± 0.0011              |
> > > | + PANDA          | 0.5214 ± 0.0068           | 0.3003 ± 0.0019              |
> > > | **+ FLAN**       | **0.5375 ± 0.0041**       | **0.2886 ± 0.0021**          |
> > >
> > >
> > > [1] Bamberger, J., Gutteridge, B., le Roux, S., Bronstein, M. M., & Dong, X. (2025). *On Measuring Long-Range Interactions in Graph Neural Networks*. ICML 2025.
> > >
> > > [2] Dwivedi, V. P., Rampášek, L., Galkin, M., Parviz, A., Wolf, G., Luu, A. T., & Beaini, D. (2022). *Long Range Graph Benchmark*. NeurIPS 2022.
> > >
> > > [3] Alon, U., & Yahav, E. (2021). *On the Bottleneck of Graph Neural Networks*. ICLR 2021.
> > >
> > > [4] Nguyen, K., et al. (2023). *Revisiting Oversmoothing and Oversquashing Using Ollivier–Ricci Curvature*. ICML 2023.
> > >
> > > [5] Wilson, J., Bechler-Speicher, M., & Velicković, P. (2024). *Cayley Graph Propagation*. LoG 2024.
> > >
> > > [6] Choi, J., Park, S., Wi, H., Cho, S.-B., & Park, N. (2024). *PANDA: Expanded Width-Aware Message Passing Beyond Rewiring*. ICML 2024.

---

> ### Author Response · Authors · 2025-11-19
> **Rebuttal part 4**
>
> **(W3) and (W6) Clarification of the correlation of Table 1**
>
>
> Thank you for this remark. We acknowledge that the strength of the correlation varies across datasets. This is expected, because the interplay between model behaviour and graph structure is not the same across datasets. The purpose of Table 1 is not to show uniformly strong correlations everywhere, but to demonstrate that, on datasets where structure governs the model’s propagation behaviour, the structural descriptors used in rewiring methods are sufficient to predict the long-range index, which justifies their effectiveness.
>
>
> ---
>
> **(W4) More details on datasets**
>
> A detailed table (See below)  summarizing these properties, grouped by dataset type (heterophilic, homophilic, long-range, node and graph-level), will be added to the appendix to clarify the structural differences across datasets and make the link between these characteristics and long-range behavior more explicit.
>
>
> | **Dataset**       | **#Nodes** | **#Edges** | **Avg(d)** | **Std(d)** | **Max(d)** | **Diam.** | **H** |
> |-------------------|-----------:|-----------:|-----------:|-----------:|-----------:|----------:|------:|
> | Cora              | 2.7k  | 5.3k  | 3.9  | 5.2   | 168  | 19   | 0.81 |
> | Citesser          | 3.3k  | 4.6k  | 2.7  | 3.4   | 99   | 28   | 0.74 |
> | Texas             | 183   | 295   | 3.22 | 7.81  | 104  | 8    | 0.11 |
> | Wisconsin         | 251   | 466   | 3.71 | 7.95  | 122  | 8    | 0.20 |
> | Cornell           | 183   | 280   | 3.06 | 7.01  | 94   | 8    | 0.13 |
> | Chameleon         | 2.3k | 31.4k | 27.60 | 46.43 | 732  | 11   | 0.24 |
> | Squirrel          | 5.2k | 198k  | 76.33 | 161.46 | 1905 | 10 | 0.22 |
> | Roman Empire      | 22.7k | 32.9k | 2.91 | 1.03 | 14 | 6824 | 0.05 |
> | Amazon-Ratings    | 24k | 93k | 7.60 | 6.00 | 132 | 46 | 0.38 |
>
> ---
>
>
> | **Dataset**        | **#Graphs** | **#Nodes (±)** | **#Edges (±)** | **Avg(d)** | **Std(d)** | **Max(deg)** | **Diam.** |
> |--------------------|------------:|----------------:|----------------:|-----------:|-----------:|--------------:|----------:|
> | MUTAG              | 188   | 17.9 (±4.6)    | 19.8 (±5.7)    | 2.2 | 0.7 | 3.0   | 8.2  |
> | PROTEINS           | 1113  | 39.1 (±45.8)   | 72.8 (±84.6)   | 3.7 | 0.9 | 5.8   | 11.6 |
> | ENZYMES            | 600   | 32.6 (±15.3)   | 62.1 (±25.5)   | 3.9 | 1.0 | 6.1   | 10.9 |
> | REDDIT             | 2000  | 429.6 (±554.1) | 497.8 (±623.0) | 2.3 | 8.9 | 217.4 | 9.7  |
> | IMDB               | 1000  | 19.8 (±10.1)   | 96.5 (±105.6)  | 8.9 | 2.8 | 18.8  | 1.9  |
> | Peptides-func      | 10873 | 151.5 (±84.1)  | 154.3 (±86.0)  | 2.0 | 0.8 | 3.0   | 57.1 |
> | Peptides-struct    | 10873 | 151.5 (±84.1)  | 154.3 (±86.0)  | 2.0 | 0.8 | 3.0   | 57.1 |
>
>
> ---
>
> **(W5) Clarity of the code**
>
> We will update the public GitHub repository with a clean version of the code, including a detailed README  and scripts corresponding to the tables and figures reported in the paper.
>
>
> ---
>
> **(W7) Clarification on “computational bottlenecks’’ and their relation to over-smoothing**
>
> Thank you for the comment. The computational bottleneck [7] refers to a limitation inherent to message passing itself: because message passing is strictly local, each node receives information only through successive hops and must therefore compress an exponentially growing number of long-range signals into a fixed-dimensional embedding. This phenomenon arises independently of the graph structure,it is a consequence of the local, iterative nature of GNNs, rather than of any specific topological substructure (in contrast with the topological bottleneck).
>
> In our opinion, this notion is distinct from over-smoothing, where representations become indistinguishable as the depth increases. Could you please clarify the connection you are referring to between computational bottlenecks and over-smoothing?
>
> [7]  Arnaiz-Rodríguez, A., & Errica, F. (2025). Oversmoothing, “Oversquashing”, Heterophily, Long-Range, and more: Demystifying Common Beliefs in Graph Machine Learning.

---

> ### Author Response · Authors · 2025-11-19
> **Rebuttal part 5**
>
> **(Q1) . Single-axis sensitivity and rank-1 correction**
>
> The sensitivity index compresses long-range demand into a single axis because the dominant error of the backbone varies monotonically with \\(p_G\\).  The rank-1 translation shifts node representations along this axis, while the diagonal reweighting rescales each feature dimension accordingly.  Together, these operations form the minimal intervention needed to fix the under-performing high \\(p_G\\) regime without affecting the rest of the representation space.
>
> ---
>
> **(Q2) Notation**
>
> Thank you for carefully pointing out these notational inconsistencies. In the updated version we will unify all symbols, remove the unnecessary bar notation, and clearly distinguish between the node-level and graph-level long-range indices.
>
> ---
>
>
> **(Q3) Interpretation of the correlation**
>
> We view this phenomenon as interesting because it reveals a link between the model and the graph structure: when a structural proxy can recover \\(p_u\\) closely, it indicates that certain topological patterns are sufficiently informative to predict a quantity that is defined from the model’s Jacobian (See also the answer to W1.4) . This helps identify which structural properties drive long-range sensitivity, and clarifies when model behaviour can be anticipated from the graph alone.
>
> ---
>
> We thank the reviewer once again for their insightful comments. We remain open to any further questions or clarifications, and we hope that the additional analyses provided here help convey the strength and relevance of our contributions.

---

> > ### Comment · Reviewer_REWK · 2025-11-27
> > **Response to authors' rebuttal**
> >
> > I thank the authors for their detailed response, which strengthen the paper and alleviate some of my concerns.
> >
> > ---
> >
> > > Although our operational cutoff aligns with the locality of one-step message passing, the empirical relationship between long-range sensitivity and predictive margin is robust to how the hops are grouped
> >
> > I'm satisfied that correlation with the margin is maintained for different cutoffs, but the index itself (along with FLAN) is the primary contribution, and your explanation doesn't quite convince me that the index is anything more than a 1-hop (or 2-, etc, depending on the cutoff) influence measurer, rather than a long-range measure. Fundamentally the index is defined in such a way that it only distinguishes inside and outside of a k-hop cutoff. I think it would be interesting to see $\rho_u^{(1, ... k)}$ plotted for varying $k$ up to the point when $\rho$ becomes negligible, to see how specific hop radii contribute to performance (though I wouldn't expect the authors to include this during this rebuttal period).
> >
> > ---
> >
> > > If the model fails to capture long-range information, the correlation between the index and the margin becomes negative (as observed in heterophilic graphs).
> >
> > But heterophily and long-range are not the same thing, and heterophilic graphs are not necessarily long-range; their range is just >1-hop. Your index seems to me better suited to measuring heterophily (i.e. lack of dependence on the 1-hop, immediate neighbors) than LR.
> >
> > ---
> >
> > > (W2) Additional experiments on long-range benchmarks.
> >
> > The additional experiments on LRGB are welcome (and should be accompanied by a discussion of its drawbacks), but I agree with reviewer XWMr that the utility of these additional results is severely limited by the fact that you use the experimental setup of the LRGB paper itself and not the one adopted by [1, 2] that fixes the major issue of basic hyperparameter tuning. As reviewer XWMr notes, the reported performance on FLAN would be more meaningful if it were compared with a reasonably tuned GCN, etc. Also, I find it odd that you would not run experiments on VOCSuperpixels from LRGB, the best candidate for a long-range task from that benchmark [1] and a node classification task.
> >
> > ---
> >
> > > (W4) More details on datasets
> >
> > I am happy to see this. I reiterate my recommendation about grouping and labelling dataset types in your other tables, for clarity, in addition to including this new table.
> >
> > ---
> >
> > > We will update the public GitHub repository with a clean version of the code, including a detailed README and scripts corresponding to the tables and figures reported in the paper.
> >
> >  I am glad to hear this, but I would personally consider reproducible code to be essential for ICLR papers at initial submission, rather than only at the camera-ready stage.
> >
> > ---
> >
> > > (W7) Clarification on “computational bottlenecks’’ and their relation to over-smoothing
> >
> > I think it is reasonable to see a similarity between the line from your paper, "signals and gradients from distant nodes attenuate through repeated local updates", and your own explanation of over-smoothing as "representations become indistinguishable as the depth increases", as greater depth means more local updates. This is a minor point overall, and I don't necessarily expect this paper to tackle the thorny issue of how over-smoothing, -squashing and vanishing gradients interrelate. But I think these behaviours (i.e. over-smoothing and vanishing gradients) are at least worth a brief discussion.
> >
> > ---
> > ### Misc
> > - You still do not mention graph Transformers, or multi-hop MPNNs, or any other architectures that explicitly address long-range interactions, other than rewiring methods
> > - W2.3 and W2.4 were not acknowledged.

---

> > > ### Author Response · Authors · 2025-11-30
> > >
> > > Thank you for the additional feedback.
> > >
> > > ---
> > >
> > >  To address your final concerns and follow your suggestion to “plot $\rho_u^{(1,\ldots,k)}$ for varying $k$ until $\rho$ becomes negligible,” we now include the cumulative distance profiles $\rho_u^{(1,\ldots,k)}$ for $k=1,\ldots,10$ in **Figures 12–15** (MUTAG, IMDB-BINARY, PROTEINS, and ENZYMES, respectively). These plots make explicit how the margin-aligned sensitivity mass is distributed across hop distances: early saturation indicates negligible contributions from larger neighborhoods, whereas saturation only after several hops suggests that a substantial share of the sensitivity is carried by multi-hop neighborhoods, i.e., beyond the one-hop component captured by our main index. Moreover, by reporting the FLAN gain curve $\Delta(k)$ on the same figures, we can identify **which hop ranges are most associated with performance improvements**, i.e., where increased captured sensitivity mass aligns with the largest gains. (All details are available in Appendix C.)
> > >
> > > ---
> > >
> > > Concerning the long-range benchmark and the multi-hop graph transformers baseline, we followed the re-tuning protocol of Tönshoff et al. (TMLR’24) on the long-range LRGB benchmarks and report a **tuned GCN baseline** together with **GCN + FLAN** over multiple seeds. Concretely, we vary (i) the use of Batch Normalization and (ii) the prediction head, replacing the linear classifier with an MLP of depth $d\in\{1,2,3\}$, while using a base learning rate $\mathrm{lr}=10^{-3}$ together with a learning-rate schedule.
> > >
> > >
> > >
> > > | Dataset          | Metric (↑/↓) | GCN (tuned)       | GCN + FLAN          | Δ |
> > > |------------------|--------------|-------------------|---------------------|---|
> > > | Peptides-func    | AP (↑)       | 0.6655 ± 0.0039   | 0.6868 ± 0.0040     | +0.0214 |
> > > | Peptides-struct  | MAE (↓)      | 0.2558 ± 0.0024   | 0.2450 ± 0.0026     | −0.0109 |
> > > | PASCALVOC-SP  | macro F1     (↑)    | 0.1792 ± 0.0039   | 0.1952 ± 0.0031     | +0.0160 |
> > >
> > > These results show that **even after tuning the base GCN to a competitive regime on long-range benchmarks**, FLAN yields **consistent improvements across runs** on both tasks.
> > > Finally, in the revised manuscript (See Appendix B), **Table 14** reports comparisons against **multi-hop and graph transformer baselines**; **GCN + FLAN achieves near state-of-the-art performance while reducing runtime by approximately 45%–95% relative to graph-transformer models**, providing additional evidence that FLAN yields practical gains beyond simply recovering an under-tuned baseline.
> > >
> > >
> > > [1] Tönshoff et al., *Where did the gap go? Reassessing the long-range graph benchmark*, TMLR 2024.
> > >
> > > ---
> > >
> > > Thank you again for all your remarks, which helped improve this work.

---

### Official Review · Reviewer_XWMr · 2025-10-28

**Soundness:** 3
**Presentation:** 3
**Contribution:** 2
**Rating:** 2
**Confidence:** 4

**Summary:**

This work revisits the problem of _over-squashing_ in GNNs, which limits the ability of distant nodes to effectively exchange information. The authors argue that the importance of long-range information is task- and node-dependent, so interventions should be adaptive rather than universal. To quantify this adaptivity, they introduce a Jacobian-based sensitivity index that measures how a node’s classification margin changes with local versus non-local perturbations. Furthermore, they find that this sensitivity can be predicted from graph structure alone. Finally, they propose FLAN, a _rewiring-free_ and _lightweight conditioning layer_ that adjusts the readout stage based on the structure-predicted sensitivity proxy.

**Strengths:**

- The motivation and presentation of this work are clear and intuitive. The idea seems novel and potentially useful.
- Various setups are used (different tasks, GNN backbones, and dataset sizes) to support the effectiveness of the method.

**Weaknesses:**

- The biggest weakness is the limited empirical evaluation of the proposed FLAN method. I appreciate that the authors are considering multiple architectures (GCN, GIN) for their experiments, but some of the datasets typically used for assessing over-squashing alleviation/long-range capabilities are not being used. For instance, the TUDataset benchmark is common, but the graphs contained in it do not have large diameters that would require long-range information propagation. A more standard benchmark for this is the Long-Range Graph Benchmark [[6], [7]]. Synthetic datasets such as Trees-NeighborsMatch [[8]] are also commonly used. Moreover, datasets such as Chameleon from WebKB have previously been criticized, and alternative datasets have been proposed for evaluation under heterophily [[9]].
- Another weakness of this work is that FLAN is a rewiring-free method, but there is no comparison against other rewiring-free baselines like PANDA [[1]], GESNs [[2]], Graph ViT/MLP-Mixer [[3]].
- If I understand correctly, the linear model is an estimator for p_u, but there is no theoretical justification or derivation explaining why this estimator is expected to be a good approximation for p_u.
- As I understand, the points in Figure 1 show different data items (nodes for Cornell, graphs for MUTAG) and their calculated p_u/m_u. Why is there so little of them, though? E.g., for Cornell, I'd expect ~100 points, and if not all are plotted, which ones exactly are plotted?
- Why do we want the estimator to be sparse? Is sparsity a way to interpret which of the 4 measures matter most?
- (Minor) Within the rewiring approaches, there are some baselines missing, such as approaches using virtual nodes or multi-hop neighborhoods [[4], [5]].
- (Very minor) Please consider saving the figures in the Appendix as PDFs/SVGs instead of images so that they will render better when zoomed.

[1]: https://arxiv.org/pdf/2406.03671
[2]: https://arxiv.org/pdf/2212.06538
[3]: https://proceedings.mlr.press/v202/he23a.html
[4]: https://arxiv.org/pdf/2201.12674
[5]: https://proceedings.mlr.press/v202/shirzad23a/shirzad23a.pdf
[6]: https://arxiv.org/abs/2206.08164
[7]: https://arxiv.org/abs/2309.00367
[8]: https://arxiv.org/abs/2006.05205
[9]: https://arxiv.org/pdf/2302.11640
[10]: https://arxiv.org/pdf/2405.19121?

**Questions:**

Please see weaknesses above.

I believe that the overall idea is interesting, and from the results shown until this moment, it does have potential. The paper is also relatively well-written. However, the experimental evaluation is very limited, and it is unclear to me if FLAN can truly alleviate over-squashing/smoothing or help with long-range dependencies in practice.

I would highly recommend that the authors expand their empirical evaluation and add at least experiments on the LRGB and some synthetic datasets, such as Trees-NeighborsMatch. Another very interesting option would be something akin to the associative recall tasks from [[10]]. I am willing to reassess my decision depending on new experiments provided in the rebuttal period. However, right now there is very little evidence that the proposed method and techniques help in practice.

[10]: https://arxiv.org/pdf/2405.19121?

---

> ### Author Response · Authors · 2025-11-19
> **Rebuttal part 1**
>
> We thank the reviewer for their careful evaluation and constructive feedback, which helped us strengthen both the clarity and the scope of the paper.
>
> **(W1, W2 and W6) Additional experiments on long-range benchmarks.**
>
> We extended our study to the Long-Range Graph Benchmark (LRGB) [2], including the Peptides-struct and Peptides-func datasets, as well as the synthetic Tree-Neighbors-Match dataset [3]. For the LRGB datasets, the notion of “margin’’ must generalize beyond standard multi-class classification. We therefore redefine it so that a larger value always indicates better predictions, regardless of the task type. These datasets include both multi-label graph classification (Peptides-func) and graph-level regression (Peptides-struct).
>
> Since regression targets in Peptides-struct are standardized (zero mean, unit variance), we define the graph-level margin as:
>
> $$
> m_G = -\log\left( \frac{1}{T}\sum_{t=1}^T |\hat{y}_t - y_t| \right), \quad m_G \in (-\infty, 0].
> $$
>
> The margin approaches $0$ when the prediction error approaches $0$, and decreases toward $-\infty$ otherwise.
>
> For Peptides-func, we define the margin as the average signed logit:
>
> $$
> m_G = \frac{1}{T} \sum_{t=1}^T (2y_t - 1)\, z_t, \quad m_G \in (-\infty, +\infty).
> $$
>
> This quantity increases when logits $z_t$ confidently align with ground-truth labels, and decreases when predictions contradict them.
>
> We report the correlation between the margin $m_G$ and the long-range capture index $p_u$ on the long-range benchmarks. Both Peptides-struct and Peptides-func exhibit a positive correlation, indicating that models perform better when they rely less on distant information. This suggests that, despite being part of a long-range benchmark, these tasks are effectively dominated by local interactions, consistent with prior work [1].
>
> | **Dataset**           | **GCN**               | **GIN**                |
> |-----------------------|-----------------------|------------------------|
> | Tree-Neighbors-Match | −0.2964 ± 0.0535      | −0.1806 ± 0.0686       |
> | Peptides-struct      | 0.3163 ± 0.0395       | 0.3343 ± 0.0329        |
> | Peptides-func        | 0.6963 ± 0.0488       | 0.5463 ± 0.0612        |
>
> We also evaluate the proposed FLAN model on Peptides-struct and Peptides-func, following the experimental protocol of [4] and [5]. We include two baselines: PANDA [6] and a virtual-node augmentation. Despite the local nature of these tasks, FLAN still improves performance: since graphs with lower long-range sensitivity achieve higher margins, adjusting embeddings according to their predicted structural index $p_G$ naturally reinforces the part of the representation space best aligned with these tasks.
>
> | **Model**        | **Peptides-func (AP ↑)** | **Peptides-struct (MAE ↓)** |
> |------------------|---------------------------|------------------------------|
> | GCN              | 0.5029 ± 0.0058           | 0.3587 ± 0.0006              |
> | + SDRF           | 0.5041 ± 0.0026           | 0.3559 ± 0.0010              |
> | + FoSR           | 0.4534 ± 0.0090           | 0.3003 ± 0.0007              |
> | + EGP            | 0.4972 ± 0.0023           | 0.3001 ± 0.0013              |
> | + CGP            | 0.5106 ± 0.0014           | 0.2931 ± 0.0006              |
> | + VN             | 0.5022 ± 0.0014           | 0.3241 ± 0.0016              |
> | + PANDA          | 0.5188 ± 0.0022           | 0.3098 ± 0.0011              |
> | **+ FLAN**       | **0.5479 ± 0.0041**       | **0.2724 ± 0.0019**          |
> |                  |                           |                              |
> | GIN              | 0.5124 ± 0.0055           | 0.3544 ± 0.0014              |
> | + SDRF           | 0.5122 ± 0.0061           | 0.3515 ± 0.0011              |
> | + FoSR           | 0.4584 ± 0.0079           | 0.3008 ± 0.0014              |
> | + EGP            | 0.4926 ± 0.0070           | 0.3034 ± 0.0027              |
> | + VN             | 0.5137 ± 0.0060           | 0.3197 ± 0.0021              |
> | + CGP            | 0.5159 ± 0.0059           | 0.2910 ± 0.0011              |
> | + PANDA          | 0.5214 ± 0.0068           | 0.3003 ± 0.0019              |
> | **+ FLAN**       | **0.5375 ± 0.0041**       | **0.2886 ± 0.0021**          |
>
>
> [1] Bamberger, J., Gutteridge, B., le Roux, S., Bronstein, M. M., & Dong, X. (2025). *On Measuring Long-Range Interactions in Graph Neural Networks*. ICML 2025.
>
> [2] Dwivedi, V. P., Rampášek, L., Galkin, M., Parviz, A., Wolf, G., Luu, A. T., & Beaini, D. (2022). *Long Range Graph Benchmark*. NeurIPS 2022.
>
> [3] Alon, U., & Yahav, E. (2021). *On the Bottleneck of Graph Neural Networks*. ICLR 2021.
>
> [4] Nguyen, K., et al. (2023). *Revisiting Oversmoothing and Oversquashing Using Ollivier–Ricci Curvature*. ICML 2023.
>
> [5] Wilson, J., Bechler-Speicher, M., & Velicković, P. (2024). *Cayley Graph Propagation*. LoG 2024.
>
> [6] Choi, J., Park, S., Wi, H., Cho, S.-B., & Park, N. (2024). *PANDA: Expanded Width-Aware Message Passing Beyond Rewiring*. ICML 2024.

---

> ### Author Response · Authors · 2025-11-19
> **Rebuttal part 2**
>
> **(W3) and (W5) Clarification on the linear model**
>
> The linear model is not intended as a theoretically derived estimator of \\(p_u\\), but rather as an interpretable structural probe : it quantifies how much of the long-range sensitivity  \\(p_u\\) is predictable solely from topology, independently of node features or training dynamics.
>
> To clarify why this probe is meaningful, we now include two simple baselines:
> (i) a degree-only predictor (min, mean and max degree), and
> (ii) a random baseline obtained by shuffling the target values.
>
> As shown in the Table below, the structural indicators widely used in rewiring methods (PageRank, Forman curvature, commute time) achieve substantially higher Test \\(R^2\\) than both baselines. The degree-only predictor performs poorly, and the random baseline yields strongly negative \\(R^2\\). This demonstrates that these structural descriptors encode the parts of the topology most associated with the model-derived \\(p_u\\), whereas degree alone is insufficient.
> This helps explain why rewiring strategies based on these metrics are often effective, while leaving room for future methods to incorporate new structural predictors.
>
>
>
> | **Dataset**       | **Structural**           | **Degree-only**         | **Random**              |
> |-------------------|---------------------------|---------------------------|---------------------------|
> | MUTAG             | 0.9900 ± 0.0057           | 0.2028 ± 0.1478           | −1.5234 ± 0.3083          |
> | PROTEINS          | 0.9395 ± 0.0150           | 0.1444 ± 0.0884           | −1.0524 ± 0.2995          |
> | ENZYMES           | 0.8721 ± 0.0411           | 0.0754 ± 0.0556           | −0.6743 ± 0.3158          |
> | Cornell           | 0.6745 ± 0.0463           | −0.0060 ± 0.0097          | −1.0397 ± 0.1161          |
> | Texas             | 0.5546 ± 0.0474           | −0.7710 ± 1.4327          | −0.9665 ± 0.4239          |
> | Wisconsin         | 0.4530 ± 0.0644           | −0.0581 ± 0.0664          | −1.1481 ± 0.2667          |
> | Roman-Empire      | 0.5196 ± 0.0713           | −0.0101 ± 0.0345          | −1.2238 ± 0.4580          |
> | Amazon-Ratings    | 0.7054 ± 0.0511           | 0.0875 ± 0.0597           | −0.9520 ± 0.1935          |
>
> A similar trend is observed when using a GIN backbone.
> We will update the table in the revised version and include a brief discussion clarifying the motivation for using a sparse linear model.
>
> ---
>
> **(W4)  Clarification on the number of points in Figure 1**
>
> In Figure 1, we only plot the test nodes  or test graphs. This explains why the number of points is smaller than the total number of data items.

---

> ### Author Response · Authors · 2025-11-19
> **Rebuttal part 3**
>
> **(Q) Attributing FLAN’s performance.**
>
> To show that FLAN focuses on the graphs that are most sensitive to long-range degradation, we examine how the margin improvement varies across different values of the long-range index $p_G$. As reported in the table below, the performance gains are concentrated on the graphs with the highest $p_G$, i.e., those whose structure is most affected by long-range dependencies. This indicates that our method specifically benefits the graphs for which message passing is most challenged by long-range effects. The table quantifies this behavior by showing how the margin improvement $\Delta m$ increases when restricting evaluation to the top 50%, 25%, and 10% most long-range-sensitive graphs.
>
> | **Dataset** | **Backbone** | **Δm (all)** | **Δm (Top50%)** | **Δm (Top25%)** | **Δm (Top10%)** |
> |-------------|--------------|--------------|------------------|------------------|------------------|
> | MUTAG       | GCN | +0.3944 | +0.1027 | +0.1101 | +0.0921 |
> |             | GIN | +0.1617 | +0.0428 | +0.1091 | +0.1547 |
> | ENZYMES     | GCN | +0.2675 | +0.2424 | +0.4382 | +0.9524 |
> |             | GIN | +0.0526 | −0.0081 | +0.3468 | +0.4252 |
> | PROTEINS    | GCN | +0.0842 | +0.1217 | +0.4729 | +1.1336 |
> |             | GIN | +0.1917 | +0.1790 | +0.4149 | +0.8718 |
> | IMDB| GCN | +0.0180 | +0.0580 | +0.0854 | +0.1258 |
> |             | GIN | +0.4964 | +0.8223 | +1.5778 | +2.4207 |
>
> We also include an ablation where FLAN is given access to the *true* backbone-derived $p_G$ instead of the predicted value (denoted **FLAN★**). The comparison shows that using the true $p_G$ leads to improvements nearly identical to those obtained with the predicted index, confirming that the benefit comes from conditioning on long-range sensitivity rather than the estimation procedure itself.
>
> | **Backbone** | **Method** | **ENZYMES**      | **IMDB**   | **MUTAG**         | **PROTEINS**      |
> |--------------|------------|------------------|--------------------|--------------------|--------------------|
> | **GCN**      | **FLAN**   | 33.8 ± 1.8       | 54.8 ± 1.6         | 81.2 ± 2.5         | 74.3 ± 1.7         |
> |              | **FLAN★**  | 34.7 ± 1.7       | 55.1 ± 1.5         | 81.6 ± 1.8         | 74.5 ± 2.1         |
> | **GIN**      | **FLAN**   | 35.8 ± 1.9       | 72.0 ± 1.3         | 81.3 ± 2.7         | 74.2 ± 1.7         |
> |              | **FLAN★**  | 36.3 ± 1.5       | 72.9 ± 1.8         | 81.1 ± 2.0         | 74.5 ± 2.3         |
>
>
> ---
>
> We thank the reviewer once again for their insightful comments. We remain open to any further questions or clarifications, and we hope that the additional analyses provided here help convey the strength and relevance of our contributions.

---

> > ### Comment · Reviewer_XWMr · 2025-11-25
> >
> > I would like to thank the authors for their rebuttal.
> >
> > * Regarding the LRGB - while I do agree with the authors' comments that the LRGB is a flawed dataset, it is still one of the few (non-synthetic) datasets that are often used when evaluating GNNs with long-range capabilities, and an acceptable proxy for comparing these models, both from a "long-range" perspective but, more importantly, from a capabilities perspective. It is unclear for me why both your baselines and your FLAN adaptations obtain results that are this poor --- a normal GCN can obtain ~0.68AP when properly tuned [1]. Regarding Neighbors-Match, I would have expected a comparison for depth v.s. accuracy when classifying the root node.
> >
> > * Regarding the new analysis on the TUDataset: while I do believe that targeting specific long-range samples is a good idea and can be interesting, I still believe that the TUDataset is generally a very limited benchmark for modern models generally. The datasets are small, they have a small diameter, and the benchmark is (mostly) saturated. I would personally even prefer more synthetic experimental results than results on the TUDataset, and that very little conclusions can be drawn from the results reported on it.
> >
> > I will keep my score and confidence. I believe that the idea is interesting and has potential, and I do encourage the authors to expand their experimental setup beyond what is reported in the paper. This work could be valuable if the experiments section would be more comprehensive. Moreover, I would suggest to the authors to further tune their baselines. For instance, specifically for the LRGB, it is very easy to reproduce results similar to the ones in [1].
> >
> > [1]: Tönshoff et al., "Where did the gap go? Reassessing the long-range graph benchmark" TMLR '24

---

> > > ### Author Response · Authors · 2025-11-26
> > >
> > > We thank the reviewer for the additional feedback.
> > >
> > >
> > > **TU datasets.**
> > > The TU datasets remain widely used for graph classification, including in evaluations of rewiring (and related) approaches for long-range/over-squashing analysis [1-6].
> > > In our work, we use the TU datasets as a controlled analysis setting to study non-local sensitivity and the effect of FLAN, rather than to claim state-of-the-art capabilities.
> > >
> > > **LRGB protocol.**
> > > For LRGB, we follow the fixed data split and evaluation protocol that is widely used in the literature (e.g., in works [2,3,4]).
> > > We agree that stronger GNN results can be obtained with more careful tuning; however, our goal here is a controlled comparison under a fixed protocol commonly used in the literature rather than maximizing the performance of each baseline.
> > >
> > > ---
> > >
> > > **More generally, we emphasize that our contributions are supported consistently across the commonly used datasets and baselines for this setting, and we further expanded the evaluation by incorporating the additional datasets suggested during the discussion. While stronger absolute numbers can sometimes be obtained with additional fine-tuning, this does not affect the clarity, validity, or originality of the proposed methodology and the empirical trends we report.**
> > >
> > >
> > >
> > >
> > > ---
> > >
> > > [1] Langzhang Liang, Fanchen Bu, Zixing Song, Zenglin Xu, Shirui Pan, Kijung Shin. *Mitigating Over-Squashing in Graph Neural Networks by Spectrum-Preserving Sparsification.* ICML 2025.
> > >
> > > [2] JJ Wilson, Maya Bechler-Speicher, Petar Veličković. *Cayley Graph Propagation.* Learning on Graphs (LoG) 2024.
> > >
> > > [3] Jeongwhan Choi, Sumin Park, Hyowon Wi, Sung-Bae Cho, Noseong Park. *PANDA: Expanded Width-Aware Message Passing Beyond Rewiring.* ICML 2024.
> > >
> > > [4] Khang Nguyen, Hieu Nong, Vinh Nguyen, Nhat Ho, Stanley Osher, Tan Nguyen. *Revisiting Over-smoothing and Over-squashing Using Ollivier-Ricci Curvature.* ICML 2023.
> > >
> > > [5] Federico Barbero, Ameya Velingker, Amin Saberi, Michael M. Bronstein, Francesco Di Giovanni. *Locality-Aware Graph Rewiring in GNNs.* ICLR 2024.
> > >
> > > [6] Adarsh Jamadandi, Celia Rubio-Madrigal, Rebekka Burkholz. *Spectral Graph Pruning Against Over-Squashing and Over-Smoothing.* NeurIPS 2024.
> > >
> > > ---

---

> > > > ### Comment · Reviewer_XWMr · 2025-11-26
> > > >
> > > > I did not suggest that SOTA results are required on the TUDataset, and I do agree that these datasets can still support an analysis. However, they are too limited to serve as the basis for a convincing empirical comparison. The same concern applies to WebKB as well as Cora and Citeseer.
> > > >
> > > > Regarding the LRGB: I agree that one can demonstrate that a modification improves results without necessarily achieving SOTA. However, in your current experiments, the results are so low that it is difficult to understand why FLAN helps. I cannot tell whether the mechanism improves the training dynamics of the GCN and recovers some performance, or whether the gains truly come from enhanced long-range capabilities. If the base GCN performed closer to a reasonably tuned GCN, and FLAN further improved that baseline, it would be much easier to evaluate the contribution of FLAN.
> > > >
> > > > Using better or more appropriate datasets and stronger baselines would significantly improve the clarity of your results. For example, if your base GCN achieved around 0.68 AP on peptides-func and FLAN improved that to 0.70 AP, it would be much easier to interpret the improvement as evidence that FLAN helps, and potentially that it helps specifically with long-range information propagation.
> > > >
> > > > To reiterate, **I do think your method is interesting and the overall approach makes sense**. However, based on the current experimental evaluation, I cannot draw clear quantitative conclusions. The method appears reasonable, but the paper is titled “When Do Distant Dependencies Matter? Diagnostics for Long-Range Propagation in GNNs.” From the reported numbers, it is impossible for me to tell whether your diagnostic or your solution actually works empirically, especially when similar performance might be achievable using a stronger model that does not have long-range capabilities.

---

> > > > > ### Author Response · Authors · 2025-11-30
> > > > >
> > > > > Thank you for continuing the discussion. To directly address your concern (“is FLAN only recovering an under-tuned baseline, or does it provide genuine long-range benefits?”), we followed the re-tuning protocol of [1] on the long-range benchmarks  and report a **tuned GCN baseline** together with **GCN + FLAN** over multiple seeds. Concretely, we vary (i) the use of Batch Normalization and (ii) the prediction head, replacing the linear classifier with an MLP of depth ($\in\{1,2,3\}$), while using a base learning rate $\mathrm{lr}=10^{-3}$ together with a learning-rate schedule.
> > > > >
> > > > >
> > > > > **LRGB (4 runs): tuned GCN vs. tuned GCN + FLAN** (mean ± std over 4 seeds)
> > > > >
> > > > > | Dataset          | Metric (↑/↓) | GCN (tuned)       | GCN + FLAN          | Δ |
> > > > > |------------------|--------------|-------------------|---------------------|---|
> > > > > | Peptides-func    | AP (↑)       | 0.6655 ± 0.0039   | 0.6868 ± 0.0040     | +0.0214 |
> > > > > | Peptides-struct  | MAE (↓)      | 0.2558 ± 0.0024   | 0.2450 ± 0.0026     | −0.0109 |
> > > > >
> > > > > These results show that **even after tuning the base GCN to a competitive regime on long-range benchmarks**, FLAN yields **consistent improvements across runs** on both tasks.
> > > > > Finally, in the revised manuscript (See Appendix B), **Table 14** reports comparisons against **multi-hop and graph transformer baselines**. **GCN + FLAN achieves near state-of-the-art performance while reducing runtime by approximately 45%–95% relative to graph-transformer models**, providing additional evidence that FLAN yields practical gains beyond simply recovering an under-tuned baseline.
> > > > >
> > > > > [1] Tönshoff et al., *Where did the gap go? Reassessing the long-range graph benchmark*, TMLR 2024.
> > > > >
> > > > > ---
> > > > >
> > > > > Thank you again for all your remarks, which helped improve this work.

---

### Official Review · Reviewer_nToS · 2025-11-01

**Soundness:** 3
**Presentation:** 3
**Contribution:** 3
**Rating:** 6
**Confidence:** 3

**Summary:**

This paper introduces a task-aligned diagnostic, the long-range capture index $p_u$, which quantifies how much margin-relevant signal is captured by one-hop information in GNN decisions. Derived from margin-aligned Jacobian sensitivities, the index reveals that reliance on distant dependencies is dataset and architecture-dependent. Crucially, $p_u$ is predictable from topology alone , with substantially higher $R^2$ than predicting margins. Building on this, the authors propose FLAN, a rewiring-free readout that conditions the classifier on the predicted $\hat{p}_u$ via per-node rescaling and bias translation. This lightweight head adaptively compensates for long-range pressure and delivers consistent gains over rewiring-based methods on both node- and graph-level benchmarks.

**Strengths:**

- Well written: Introduces a novel, task-aligned, and interpretable diagnostic, the long-range capture index, derived from margin-aligned sensitivities. This is a significant contribution to understanding the computational bottleneck in GNNs.
- The proposed FLAN layer is a minimal, parameter-efficient, and effective intervention that improves performance without altering the graph topology or increasing the model's depth.
- Strong empirical results and efficiency

**Weaknesses:**

- Computing the ground-truth sensitivity indices $p_u$ and $\rho_G$ relies on a fully trained GNN, access to labels (for margin computation), and node features (for Jacobian sensitivities), which limits the diagnostic’s applicability in label-scarce or feature-limited settings.
- On node classification, especially small heterophilous datasets, the structure-only proxy for $\hat p_u$ exhibits high variance in $R^2$, and the corresponding gains from FLAN are less stable across seeds and backbones.
- The Jacobian-based diagnostic may be computationally costly and sensitive to training noise or feature scaling on very large graphs; scalability and numerical stability are not thoroughly characterized.
- Predicting $\hat p_u$ from topology depends on a specific set of indicators (e.g., PageRank, curvature, commute time); robustness to alternative features and distribution shift is underexplored.
- FLAN modifies only the readout (per-node rescaling and bias translation) while freezing the encoder; if oversquashing stems from insufficient message-passing capacity, benefits may plateau without encoder or topology changes.
- The aggregation from node-level $p_u$ to graph-level $\rho_G$ is not deeply ablated; results could depend on choices of normalization or weighting across nodes/graphs.
- Parity with rewiring baselines (compute budget, hyperparameter tuning, and training schedules) is not fully detailed, leaving room for comparison bias.

**Questions:**

Please see the weaknesses. Also, please report FLAN performance when conditioned on (i) the oracle index $p_u$ computed post-training with labels and Jacobians, versus (ii) the structure-only proxy $\hat p_u$ computed from topology. Include mean $\pm$ std across seeds and the absolute/relative delta between the two conditions on both node- and graph-level benchmarks.

---

> ### Author Response · Authors · 2025-11-21
> **Rebuttal part 1**
>
> We thank the reviewer for their careful evaluation and constructive feedback, which helped us strengthen both the clarity and the scope of the paper.
>
> **(W1) Applicability of the diagnostic**
>
> Because our goal is to analyse how margins behave under local vs. long-range sensitivity, computing \\(p_u\\) and \\(\rho_G\\) necessarily requires to have access to labeled data.. While this limits its applicability in label-scarce settings, this diagnostic is explicitly designed as an a-posteriori tool for understanding model behaviour, and exploring label-free or self-supervised analogues is a natural direction for future work.
>
> ---
>
> **(W2) Variability on small heterophilous node-classification datasets**
>
> We agree that small heterophilous node-classification datasets are known to exhibit high variance: their limited size amplifies sensitivity to structural noise and model initialization, which affects the stability of the structure-only proxy for \\(p_u\\). Note that this does not contradict our findings, but simply reflects that long-range behaviour is harder to recover from topology alone on very small graphs, which also explains why FLAN’s gains are less stable in this regime.
>
> ---
>
> **(W3) On the Computing the ground-truth sensitivity.**
>
> For large graphs, we control the cost of computing the long-range index by using a random sample Jacobian estimation, as done in [1], which makes the diagnostic scalable and stable in practice.
>
> [1] Bamberger, J., Gutteridge, B., Le Roux, S., Bronstein, M. M., & Dong, X. (2025). *On Measuring Long-Range Interactions in Graph Neural Networks*. ICML 2025.
>
> ---
>
> **(W4) Predicting \\(\hat p_u\\) from topology**
>
> Our goal goes beyond predicting \\(p_u\\), we aim to understand when long-range sensitivity defined from the model’s Jacobian, can be anticipated from graph structure itself. The fact that structural descriptors commonly used in rewiring (PageRank, Forman curvature, commute-time resistance) can approximate the index shows that part of the long-range behaviour of GNNs is indeed governed by specific structural properties. At the same time, the proxy does not replace the index: it works only because the index provides a principled, task-aligned target.
> Importantly, simple local statistics such as min/mean/max degree fail to predict \\(p_u\\), and shuffled features perform even worse (see table below). This confirms that the index captures information that is not accessible to local structural measures. These results help explain, in general, why rewiring strategies based on these geometric descriptors are often effective, while also leaving room for future approaches to explore new, more expressive structural predictors.
>
> | **Dataset**       | **Structural**           | **Degree-only**         | **Random**              |
> |-------------------|---------------------------|---------------------------|---------------------------|
> | MUTAG             | 0.9900 ± 0.0057           | 0.2028 ± 0.1478           | −1.5234 ± 0.3083          |
> | PROTEINS          | 0.9395 ± 0.0150           | 0.1444 ± 0.0884           | −1.0524 ± 0.2995          |
> | ENZYMES           | 0.8721 ± 0.0411           | 0.0754 ± 0.0556           | −0.6743 ± 0.3158          |
> | Cornell           | 0.6745 ± 0.0463           | −0.0060 ± 0.0097          | −1.0397 ± 0.1161          |
> | Texas             | 0.5546 ± 0.0474           | −0.7710 ± 1.4327          | −0.9665 ± 0.4239          |
> | Wisconsin         | 0.4530 ± 0.0644           | −0.0581 ± 0.0664          | −1.1481 ± 0.2667          |
> | Roman-Empire      | 0.5196 ± 0.0713           | −0.0101 ± 0.0345          | −1.2238 ± 0.4580          |
> | Amazon-Ratings    | 0.7054 ± 0.0511           | 0.0875 ± 0.0597           | −0.9520 ± 0.1935          |
>
> A similar trend is observed when using a GIN backbone.
> We will update the table in the revised version and clarify the motivation for using a sparse linear model.

---

> ### Author Response · Authors · 2025-11-21
> **Rebuttal part 2**
>
> **(W5) FLAN’s effect on long-range sensitivity**
>
> This is correct: FLAN does not expand message passing capacity, so it cannot remove oversquashing on its own. Its role is complementary, as shown in the figures provided in the Appendix ( Figure 5, 6 and 7) : it specifically corrects how the model weights long-range sensitivity at readout, consistently improving the under-performing high $\rho_G$ regime even when the encoder and topology remain fixed.
>
> Additionally, to show that FLAN focuses on graphs that are more sensitive to long-range degradation, we examine how the margin improvement varies across different values of the long-range index $\rho_G$. As reported in the table below, the performance gains are concentrated on the graphs with the highest $\rho_G$, i.e., those whose structure is most affected by long-range dependencies. This indicates that our method specifically benefits the graphs for which message passing is most challenged by long-range effects. The table quantifies this behavior by showing how the margin improvement $\Delta marge$ increases when restricting evaluation to the top 50%, 25%, and 10% most long-range-sensitive graphs.
>
> | **Dataset** | **Backbone** | **Δm (all)** | **Δm (Top50%)** | **Δm (Top25%)** | **Δm (Top10%)** |
> |-------------|--------------|--------------|------------------|------------------|------------------|
> | MUTAG       | GCN | +0.3944 | +0.1027 | +0.1101 | +0.0921 |
> |             | GIN | +0.1617 | +0.0428 | +0.1091 | +0.1547 |
> | ENZYMES     | GCN | +0.2675 | +0.2424 | +0.4382 | +0.9524 |
> |             | GIN | +0.0526 | −0.0081 | +0.3468 | +0.4252 |
> | PROTEINS    | GCN | +0.0842 | +0.1217 | +0.4729 | +1.1336 |
> |             | GIN | +0.1917 | +0.1790 | +0.4149 | +0.8718 |
> | IMDB| GCN | +0.0180 | +0.0580 | +0.0854 | +0.1258 |
> |             | GIN | +0.4964 | +0.8223 | +1.5778 | +2.4207 |
>
>
> ---
>
> **(W6)] Aggregation and the role of the node-level**
>
> Thank you for the question. For graph classification tasks, the margin is defined only at the graph level, so all nodes of a given graph share the same margin $\ m_G \$ . What differs across nodes is their influence on that margin, captured by the Jacobian-based sensitivities.
> Because the margin does not vary across nodes, there is no ambiguity about normalization or weighting: only the node-level sensitivities change, not the margin itself. We will clarify this in the updated version.
>
>
> ---
>
> **(W7) Parity with rewiring baselines**
>
> Thank you for the comment. We ensured parity with all rewiring baselines along the three dimensions mentioned: compute budget, hyperparameter tuning, and training schedules. All methods were run under the same training protocol provided by [2] for fair comparison (e.g., learning rate 1e−3, hidden dimension 64, 4 layers).
> Importantly, rewiring methods typically involve several hyperparameters (e.g., the number of edges to remove/add, thresholds, and sparsification ratios). In contrast, FLAN introduces no additional hyperparameters: the backbone encoder is frozen, and FLAN only learns a per-dimension rescaling. Consequently, the compute budget, training schedule, and hyperparameter tuning are identical to those of the baseline GNN. We will state this explicitly in the revised version.
>
> [2] Langzhang Liang, Fanchen Bu, Zixing Song, Zenglin Xu, Shirui Pan, and Kijung Shin.  Mitigating Over-Squashing in Graph Neural Networks by Spectrum-Preserving Sparsification. ICML 2025.
>
> Additionally, in the Appendix, we compare FLAN’s processing runtime against graph–rewiring baselines. The reported times include (i) Jacobian–margin evaluation, (ii) computation of structural indicators, and (iii) Lasso fitting for  \\(\hat{p}_G \\). On average across datasets, our method is $10$–$1000 \times$ faster than curvature-based rewiring, spectral-gap-based rewiring, and resistance-based rewiring.
>
> ---
>
> We thank the reviewer once again for their insightful comments. We remain open to any further questions or clarifications, and we hope that the additional analyses provided here help convey the strength and relevance of our contributions.

---

### Official Review · Reviewer_Btf2 · 2025-11-01

**Soundness:** 2
**Presentation:** 3
**Contribution:** 3
**Rating:** 4
**Confidence:** 3

**Summary:**

This paper investigates the role of long-range interactions in graph neural networks (GNNs) for both node-level and graph-level tasks. It introduces a Jacobian-based metric that quantifies the contribution of one-hop interactions to the decision margin, relative to interactions spanning two or more hops. The authors show that this measure can largely be predicted from the graph structure alone. Finally, the paper proposes a rewiring-free layer designed to enhance long-range information propagation.

**Strengths:**

1. **Clarity** The paper is well written.
2. **Significance of the problem** The problem of evaluating the importance of long range interactions is timeline and of interest.
3. **Novelty** The results that topology alone can be used to predict long-rangedness is novel and interesting.
4. **Scope** The paper has a large scope, covering both diagnostic tool and solution.

**Weaknesses:**

1. **Limited coverage of broader literature**
The Background and Related Work section provides an extensive overview of over-squashing and rewiring but does not sufficiently discuss related research on long-range interactions beyond over-squashing (e.g., [1, 2]). In particular, a comparison clarifying how the use of the Jacobian in this paper differs from [2] would help better situate the work within the broader literature.

2. **Weaknesses of the long-range capture index**
See Questions 1–3 below. A deeper discussion of the design choices underlying the long-range capture index—and the potential limitations or assumptions they entail—would further strengthen the paper.

3. **Lack of evaluation on explicitly long-range tasks**
The experiments focus on tasks that are not known to require long-range reasoning. Including an evaluation on a task with known long-range dependencies, even if synthetic, would provide stronger evidence supporting the proposed method.

_________

**References:**

[1] Dwivedi, V. P., Rampášek, L., Galkin, M., Parviz, A., Wolf, G., Luu, A. T., & Beaini, D. (2022). Long range graph benchmark. Advances in Neural Information Processing Systems, 35, 22326-22340.


[2] Bamberger, J., Gutteridge, B., le Roux, S., Bronstein, M. M., & Dong, X. (2025). On Measuring Long-Range Interactions in Graph Neural Networks. In Proceedings of the 42nd International Conference on Machine Learning.

**Questions:**

1. **Definition of the graph-level margin.**
How exactly is the graph-level margin ​defined for a node u (Equation 6)? The Jacobian of the graph-level output is a vector indexed solely by the nodes, so I do not understand how it can be binned by distances as in Equation 3.

2. **Exclusion of self-node influence.**
The long-range capture index ignores the self-node influence. Why are the nodes themselves excluded in Equation 4?

3. **Definition of short- vs. long-range interactions.**
By defining the long-range capture index as the fraction captured by the one-hop neighborhood—and considering a margin “long-range” when this number is low—you implicitly assume that a two-hop interaction is long-range. However, one could argue that a two-hop interaction might still be considered short-range. Why do you only consider one-hop neighbors as short-range interactions?

4. **Experimental setup**
For node-level tasks, GNNs of only two layers were considered to evaluate the correlation between the long-range capture index (Fig 1 and Fig 2). One may argue that these networks are too shallow to process long-range information. Do you expect the results to hold for deeper networks?

5. **Error bars in Figure 2.**
Could you please report error bars for the correlations in Figure 2?

6. **Interpretability of Table 1.**
In Table 1, it is difficult to interpret what the reported numbers represent. Could you include a simple baseline for comparison, perhaps a random baseline or one based on node degrees?

7. In Table 1 and Figure 3, are the coefficients shown for a single run, or are they the mean coefficients computed over the 20 runs (as in Figure 2)?

8. **Attributing FLAN’s performance.** Would it be possible to attribute FLAN’s improved performance to its ability to better model long-range interactions? For instance, could you compute the one-hop index of the trained FLAN model and compare it with that of the backbone model?

---

> ### Author Response · Authors · 2025-11-19
> **Rebuttal Part 1**
>
> We thank the reviewer for their careful evaluation and constructive feedback, which helped us strengthen both the clarity and the scope of the paper.
>
> **(W1) Limited coverage of broader literature and clarifying how the use of the Jacobian in this paper differs from [1].**
>
> Our diagnostic index measures a task-aligned, margin-aware form of long-range sensitivity, whereas the range measure in [1] is operator-aligned and intentionally task-agnostic. Their metric quantifies how far Jacobian/Hessian influence can propagate across the graph, but it does not indicate whether such influence is relevant to the decision boundary. In contrast, our index isolates the share of margin-relevant sensitivity captured at one hop versus longer ranges, which allows us to determine when long-range propagation helps or hurts performance.
> We will clarify this distinction in the revised version, including a brief discussion comparing the intent, scope, and interpretability of both metrics in the context of long-range behavior.
>
> [1] Bamberger, J., Gutteridge, B., Le Roux, S., Bronstein, M. M., & Dong, X. (2025). *On Measuring Long-Range Interactions in Graph Neural Networks*. ICML 2025.
>
> ---
>
> **(Q1) Definition and clarification of the graph-level margin.**
>
> Thank you for the question. In graph classification, the margin is global: all nodes share the same graph-level margin $m_G$. Our graph-level index $\rho_G$ aggregates the node-wise Jacobian contributions according to their structural role (1-hop vs.\ $\ge 2$-hop regions). This makes $\rho_G$ meaningful even though the prediction is global, as it reveals how the model distributes its sensitivity across the graph. We will clarify this point in the updated version.
>
> ---
>
> **(Q2) Exclusion of self-node influence for the capture index.**
>
> Thank you for the remark. Empirically, we observed that self-node Jacobian contributions are negligible compared to interactions with other nodes. For this reason, we excluded self-loops in Equation 4. We will clarify this choice in the updated version of the paper.
>
> ---
>
> **(Q3) Definition of short vs. long-range interactions.**
>
> While alternative choices for the boundary between short and long-range interactions are possible, our results do not depend on selecting the distance-$1$ bin as the short-range component. In practice, redefining the cutoff for example using distance $(1,2)$ or distances $(1,2,3)$ as the short-range part changes the absolute values of the index but leaves its correlation with the margin $m_u$ essentially unchanged.
>
> This stability arises because the relative ordering of nodes according to their distance-binned sensitivity distribution $\{S_u(k)\}$ is highly consistent across hop definitions. Thus, although our operational cutoff aligns with the locality of one-step message passing, the empirical relationship between long-range sensitivity and predictive margin is robust to how the hops are grouped.
>
> For illustration, the table below reports the correlations obtained with a GCN backbone on graph classification datasets:
>
> | **Model & Dataset** | **Hop Range**       | **Mean** | **Std** |
> |---------------------|----------------------|----------|---------|
> | **GCN — MUTAG**     | 1-hop only           | -0.4918  | 0.0876  |
> |                     | 1–2 hops             | -0.4440  | 0.0710  |
> | **GCN — PROTEINS**  | 1-hop only           | -0.1809  | 0.0550  |
> |                     | 1–2 hops             | -0.1676  | 0.0420  |
> | **GCN — ENZYMES**   | 1-hop only           | -0.2216  | 0.1715  |
> |                     | 1–2 hops             | -0.1559  | 0.1990  |
> | **GIN — MUTAG**     | 1-hop only           | -0.4963  | 0.0968  |
> |                     | 1–2 hops             | -0.4328  | 0.1033  |
> | **GIN — PROTEINS**  | 1-hop only           | -0.1479  | 0.0649  |
> |                     | 1–2 hops             | -0.1134  | 0.0579  |
> | **GIN — ENZYMES**   | 1-hop only           | -0.2303  | 0.1824  |
> |                     | 1–2 hops             | -0.0638  | 0.1731  |
>
> Note that even though the absolute values of the long-range index vary across hop definitions (e.g., on MUTAG the average values are approximately $\rho_u^{(1)} \approx 0.17$, $\rho_u^{(1,2)} \approx 0.34$, and $\rho_u^{(1,2,3)} \approx 0.49$), the correlation with the margin remains stable.

---

> > ### Author Response · Authors · 2025-11-19
> > **Rebuttal Part 2**
> >
> > **(W3) Additional experiments on long-range benchmarks.**
> >
> > We extended our study to the Long-Range Graph Benchmark (LRGB) [2], including the Peptides-struct and Peptides-func datasets, as well as the synthetic Tree-Neighbors-Match dataset [3]. For the LRGB datasets, the notion of “margin’’ must generalize beyond standard multi-class classification. We therefore redefine it so that a larger value always indicates better predictions, regardless of the task type. These datasets include both multi-label graph classification (Peptides-func) and graph-level regression (Peptides-struct).
> >
> > Since regression targets in Peptides-struct are standardized (zero mean, unit variance), we define the graph-level margin as:
> >
> > $$
> > m_G = -\log\left( \frac{1}{T}\sum_{t=1}^T |\hat{y}_t - y_t| \right), \quad m_G \in (-\infty, 0].
> > $$
> >
> > The margin approaches $0$ when the prediction error approaches $0$, and decreases toward $-\infty$ otherwise.
> >
> > For Peptides-func, we define the margin as the average signed logit:
> >
> > $$
> > m_G = \frac{1}{T} \sum_{t=1}^T (2y_t - 1)\, z_t, \quad m_G \in (-\infty, +\infty).
> > $$
> >
> > This quantity increases when logits $z_t$ confidently align with ground-truth labels, and decreases when predictions contradict them.
> >
> > We report the correlation between the margin $m_G$ and the long-range capture index $p_u$ on the long-range benchmarks. Both Peptides-struct and Peptides-func exhibit a positive correlation, indicating that models perform better when they rely less on distant information. This suggests that, despite being part of a long-range benchmark, these tasks are effectively dominated by local interactions, consistent with prior work [1].
> >
> > | **Dataset**           | **GCN**               | **GIN**                |
> > |-----------------------|-----------------------|------------------------|
> > | Tree-Neighbors-Match | −0.2964 ± 0.0535      | −0.1806 ± 0.0686       |
> > | Peptides-struct      | 0.3163 ± 0.0395       | 0.3343 ± 0.0329        |
> > | Peptides-func        | 0.6963 ± 0.0488       | 0.5463 ± 0.0612        |
> >
> > We also evaluate the proposed FLAN model on Peptides-struct and Peptides-func, following the experimental protocol of [4] and [5]. We include two baselines: PANDA [6] and a virtual-node augmentation. Despite the local nature of these tasks, FLAN still improves performance: since graphs with lower long-range sensitivity achieve higher margins, adjusting embeddings according to their predicted structural index $p_G$ naturally reinforces the part of the representation space best aligned with these tasks.
> >
> > | **Model**        | **Peptides-func (AP ↑)** | **Peptides-struct (MAE ↓)** |
> > |------------------|---------------------------|------------------------------|
> > | GCN              | 0.5029 ± 0.0058           | 0.3587 ± 0.0006              |
> > | + SDRF           | 0.5041 ± 0.0026           | 0.3559 ± 0.0010              |
> > | + FoSR           | 0.4534 ± 0.0090           | 0.3003 ± 0.0007              |
> > | + EGP            | 0.4972 ± 0.0023           | 0.3001 ± 0.0013              |
> > | + CGP            | 0.5106 ± 0.0014           | 0.2931 ± 0.0006              |
> > | + VN             | 0.5022 ± 0.0014           | 0.3241 ± 0.0016              |
> > | + PANDA          | 0.5188 ± 0.0022           | 0.3098 ± 0.0011              |
> > | **+ FLAN**       | **0.5479 ± 0.0041**       | **0.2724 ± 0.0019**          |
> > |                  |                           |                              |
> > | GIN              | 0.5124 ± 0.0055           | 0.3544 ± 0.0014              |
> > | + SDRF           | 0.5122 ± 0.0061           | 0.3515 ± 0.0011              |
> > | + FoSR           | 0.4584 ± 0.0079           | 0.3008 ± 0.0014              |
> > | + EGP            | 0.4926 ± 0.0070           | 0.3034 ± 0.0027              |
> > | + VN             | 0.5137 ± 0.0060           | 0.3197 ± 0.0021              |
> > | + CGP            | 0.5159 ± 0.0059           | 0.2910 ± 0.0011              |
> > | + PANDA          | 0.5214 ± 0.0068           | 0.3003 ± 0.0019              |
> > | **+ FLAN**       | **0.5375 ± 0.0041**       | **0.2886 ± 0.0021**          |
> >
> > **References**
> >
> > [1] Bamberger, J., Gutteridge, B., le Roux, S., Bronstein, M. M., & Dong, X. (2025). *On Measuring Long-Range Interactions in Graph Neural Networks*. ICML 2025.
> >
> > [2] Dwivedi, V. P., Rampášek, L., Galkin, M., Parviz, A., Wolf, G., Luu, A. T., & Beaini, D. (2022). *Long Range Graph Benchmark*. NeurIPS 2022.
> >
> > [3] Alon, U., & Yahav, E. (2021). *On the Bottleneck of Graph Neural Networks*. ICLR 2021.
> >
> > [4] Nguyen, K., et al. (2023). *Revisiting Oversmoothing and Oversquashing Using Ollivier–Ricci Curvature*. ICML 2023.
> >
> > [5] Wilson, J., Bechler-Speicher, M., & Velicković, P. (2024). *Cayley Graph Propagation*. LoG 2024.
> >
> > [6] Choi, J., Park, S., Wi, H., Cho, S.-B., & Park, N. (2024). *PANDA: Expanded Width-Aware Message Passing Beyond Rewiring*. ICML 2024.

---

> ### Author Response · Authors · 2025-11-19
> **Rebuttal Part 3**
>
> **(Q4) Experimental setup for node-level tasks with deeper GNN models.**
>
> To address this concern, we repeated the analysis using deeper 4-layer GCN, GAT, and GIN backbones, and observed that the correlation between the node margin $m_u$ and the long-range capture index $\rho_u^{(1)}$ remains consistent across architectures and depths. This confirms that the phenomenon is robust to increasing network depth. The results are reported below.
>
> | **Dataset**       | **GCN: ρ⁽¹⁾**              | **GAT: ρ⁽¹⁾**              | **GIN: ρ⁽¹⁾**              |
> |-------------------|----------------------------|-----------------------------|-----------------------------|
> | Chameleon         | −0.3841 ± 0.0526          | −0.3465 ± 0.0633           | −0.1707 ± 0.1272           |
> | Squirrel          | −0.3099 ± 0.0644          | −0.2205 ± 0.0800           | −0.2617 ± 0.1006           |
> | Texas             | −0.5350 ± 0.1086          | −0.3248 ± 0.1353           | −0.3929 ± 0.1465           |
> | Cornell           | −0.4430 ± 0.1024          | −0.3680 ± 0.1467           | −0.1322 ± 0.1826           |
> | Wisconsin         | −0.4696 ± 0.1147          | −0.3045 ± 0.0842           | −0.0676 ± 0.1424           |
> | Cora              |  0.1910 ± 0.0396          |  0.1595 ± 0.0431           |  0.1372 ± 0.0303           |
> | Citeseer          | −0.0063 ± 0.0466          |  0.0190 ± 0.0316           |  0.0176 ± 0.0279           |
> | Amazon-Ratings    | −0.1799 ± 0.0531          | −0.0850 ± 0.0645           | −0.1408 ± 0.0565           |
> | Roman-Empire      | −0.1811 ± 0.0590          | −0.0342 ± 0.0609           | −0.2034 ± 0.0857           |
>
>  We will include a brief discussion of these results in the Appendix of the revised version.
>
> ---
>
> **(Q5) Error bars in Figure 2.**
>
> We agree that including error bars improves the readability of the results. In the revised version, we will add error bars corresponding to the variability across random seeds. For reference, the table below reports the mean ± standard deviation of the correlation between the node margin $m_u$ and the long-range index $p_u$.
>
> | **Dataset**        | **GCN**                | **GAT**                | **GIN**                |
> |--------------------|------------------------|-------------------------|-------------------------|
> | Texas              | −0.4828 ± 0.1375       | −0.4728 ± 0.1427        |  0.1579 ± 0.1636        |
> | Cornell            | −0.3555 ± 0.1071       | −0.4506 ± 0.1133        |  0.0590 ± 0.2079        |
> | Wisconsin          | −0.4478 ± 0.1194       | −0.4143 ± 0.1144        |  0.1860 ± 0.1393        |
> | Chameleon          | −0.4621 ± 0.0225       | −0.4558 ± 0.0538        | −0.0192 ± 0.0523        |
> | Squirrel           | −0.2250 ± 0.0100       | −0.2800 ± 0.0415        | −0.0436 ± 0.0532        |
> | Amazon-Ratings     | −0.2058 ± 0.0716       | −0.1948 ± 0.0637        |  0.0351 ± 0.0570        |
> | Roman-Empire       | −0.1151 ± 0.0557       | −0.1561 ± 0.1088        | −0.0487 ± 0.0642        |
> | Cora               |  0.1305 ± 0.0420       |  0.1112 ± 0.0339        |  0.0003 ± 0.0572        |
> | Citeseer           | −0.0820 ± 0.0301       | −0.0741 ± 0.0663        | −0.1085 ± 0.0702        |
> | Mutag              | −0.4941 ± 0.0962       | −0.5290 ± 0.0772        | −0.4870 ± 0.0810        |
> | Proteins           | −0.1954 ± 0.0775       |  0.1737 ± 0.0795        |  0.1473 ± 0.1233        |
> | Enzymes            |  0.0511 ± 0.1242       |  0.0111 ± 0.1185        |  0.0320 ± 0.1256        |

---

> ### Author Response · Authors · 2025-11-19
> **Rebuttal Part 4**
>
> **(Q6) Interpretability of Table 1.**
>
> We thank the reviewer for the suggestion. To improve interpretability, we now include two simple baselines: (i) a degree baseline (using minimum, mean, and maximum degree), and (ii) a random baseline obtained by shuffling the target values. As shown in the Table below, the structural measures widely used in graph rewiring methods consistently achieve very high Test $R^2$ values when using a GCN backbone, whereas the degree-only predictor performs poorly and the random baseline yields strongly negative $R^2$. This indicates that the structural descriptors leveraged by rewiring methods capture the aspects of graph topology most related to the model-derived $p_u$, whereas degree alone is insufficient to explain the long-range sensitivity measured by our index. We will update the table accordingly and include a brief discussion to improve its interpretability in the updated version.
>
> | **Dataset**  | **Structural**     | **Degree-only**        | **Random**        |
> |--------------------|---------------------------|---------------------------|--------------------------|
> | MUTAG              | 0.9900 ± 0.0057           | 0.2028 ± 0.1478           | −1.5234 ± 0.3083          |
> | PROTEINS           | 0.9395 ± 0.0150           | 0.1444 ± 0.0884           | −1.0524 ± 0.2995          |
> | ENZYMES            | 0.8721 ± 0.0411           | 0.0754 ± 0.0556           | −0.6743 ± 0.3158          |
> | Cornell            | 0.6745 ± 0.0463           | −0.0060 ± 0.0097          | −1.0397 ± 0.1161          |
> | Texas              | 0.5546 ± 0.0474           | −0.7710 ± 1.4327          | −0.9665 ± 0.4239          |
> | Wisconsin          | 0.4530 ± 0.0644           | −0.0581 ± 0.0664          | −1.1481 ± 0.2667          |
> | Roman-Empire       | 0.5196 ± 0.0713           | −0.0101 ± 0.0345          | −1.2238 ± 0.4580          |
> | Amazon-Ratings     | 0.7054 ± 0.0511           | 0.0875 ± 0.0597           | −0.9520 ± 0.1935          |
>
> A similar trend is observed when using a GIN backbone.
>
> ---
>
> **(Q7) Clarification in Table 1 and Figure 3.**
>
> The coefficients in Table 1 and Figure 3 correspond to the mean values over 20 independent runs, consistent with Figure 2. We will state this explicitly in the updated version.
>
> ---
>
> **(Q8) Attributing FLAN’s performance.**
>
> To show that FLAN focuses on the graphs that are most sensitive to long-range degradation, we examine how the margin improvement varies across different values of the long-range index $p_G$. As reported in the table below, the performance gains are concentrated on the graphs with the highest $p_G$, i.e., those whose structure is most affected by long-range dependencies. This indicates that our method specifically benefits the graphs for which message passing is most challenged by long-range effects. The table quantifies this behavior by showing how the margin improvement $\Delta m$ increases when restricting evaluation to the top 50%, 25%, and 10% most long-range-sensitive graphs.
>
> | **Dataset** | **Backbone** | **Δm (all)** | **Δm (Top50%)** | **Δm (Top25%)** | **Δm (Top10%)** |
> |-------------|--------------|--------------|------------------|------------------|------------------|
> | MUTAG       | GCN | +0.3944 | +0.1027 | +0.1101 | +0.0921 |
> |             | GIN | +0.1617 | +0.0428 | +0.1091 | +0.1547 |
> | ENZYMES     | GCN | +0.2675 | +0.2424 | +0.4382 | +0.9524 |
> |             | GIN | +0.0526 | −0.0081 | +0.3468 | +0.4252 |
> | PROTEINS    | GCN | +0.0842 | +0.1217 | +0.4729 | +1.1336 |
> |             | GIN | +0.1917 | +0.1790 | +0.4149 | +0.8718 |
> | IMDB| GCN | +0.0180 | +0.0580 | +0.0854 | +0.1258 |
> |             | GIN | +0.4964 | +0.8223 | +1.5778 | +2.4207 |
>
> We also include an ablation where FLAN is given access to the *true* backbone-derived $p_G$ instead of the predicted value (denoted **FLAN★**). The comparison shows that using the true $p_G$ leads to improvements nearly identical to those obtained with the predicted index, confirming that the benefit comes from conditioning on long-range sensitivity rather than the estimation procedure itself.
>
> | **Backbone** | **Method** | **ENZYMES**      | **IMDB**   | **MUTAG**         | **PROTEINS**      |
> |--------------|------------|------------------|--------------------|--------------------|--------------------|
> | **GCN**      | **FLAN**   | 33.8 ± 1.8       | 54.8 ± 1.6         | 81.2 ± 2.5         | 74.3 ± 1.7         |
> |              | **FLAN★**  | 34.7 ± 1.7       | 55.1 ± 1.5         | 81.6 ± 1.8         | 74.5 ± 2.1         |
> | **GIN**      | **FLAN**   | 35.8 ± 1.9       | 72.0 ± 1.3         | 81.3 ± 2.7         | 74.2 ± 1.7         |
> |              | **FLAN★**  | 36.3 ± 1.5       | 72.9 ± 1.8         | 81.1 ± 2.0         | 74.5 ± 2.3         |
>
> ---
>
> We thank the reviewer once again for their insightful comments. We remain open to any further questions or clarifications, and we hope that the additional analyses provided here help convey the strength and relevance of our contributions.

---

> > ### Comment · Reviewer_Btf2 · 2025-11-24
> >
> > I thank the authors for their thorough rebuttal. The added analysis and experimental results strengthen the paper. Almost all my concerns are addressed.
> >
> > **Graph-level margin.** My remaining concern is about the definition of the long range capture index for graph-level tasks. I fail to understand the given explanation for the graph-level margin. Do you mean that you treat the graph-level task as a node-level task where all nodes have the same label, and then apply the node-level definition there?

---

> ### Author Response · Authors · 2025-11-24
> **Rebuttal clarification on graph-level margin.**
>
> Thank you for raising this point. For graph classification, we define a graph-level margin \\(m_G\\) from pooled graph logits. We then compute Jacobian sensitivities of this single scalar margin w.r.t. each node's input features, so all nodes share the same target margin \\(m_G\\) while having different gradients.
>
> This does not turn the task into node classification (no node labels are created); the shared margin is only used to decompose margin-relevant sensitivity by hop distance before averaging to obtain \\(\rho_G \\).
> We will clarify this in the update version.
> Thank you again.

---

> > ### Comment · Reviewer_Btf2 · 2025-11-24
> >
> > Thank you for the reply. If I understand correctly, once the margin and gradients are computed you obtain a vector indexed by the nodes and feature, which following the notation in section 3 we can denote by $J^G_{s, g}$ where s is a node and g is a feature. However unlike for the node-level task the Jacobian is not indexed by a target node (denoted u in equations 1-3). Without this index, I do not understand how you can define the "distance-binned aggregation" as done in Equation 3. Having a target node u is essential in order to sum over nodes s such that D(s, u)=k. In the graph-level case u is replaced by the graph G, in which case I cannot make sense of D(s, G)=k.
> >
> > Could you perhaps walk me through the exact equations in order to  compute $\rho_{u, g}$ for a node $u$ in the graph-level index definition (Equation 6)?

---

> ### Author Response · Authors · 2025-11-24
>
> Thank you for the clarification request. In the graph-level setting, the Jacobian target is graph-level, while distance binning is still performed around each node.
>
> Let pooled graph logits be \(z_G\). We define a scalar graph margin
> \\( m_G = z_G[y_G] - \max_{c\neq y_G} z_G[c]. \\)
>
> **1) Graph-level Jacobian per node.**
> For every node \(s\), we compute the Jacobian of this *single* margin with respect to node input features:
> \\( J_s = \frac{\partial m_G}{\partial x_s} \in \mathbb{R}^F, \qquad a_s = \|J_s\|\\)
> Thus we obtain one margin-aware importance score \(a_s\) per node.
>
> **2) Distance-binned aggregation around a reference node \(u\).**
> Although the margin is graph-level, we choose a node \(u\) only as a reference for hop-distance binning :
> \\( b_u(k) = \sum_{s:\,D(s,u)=k} a_s. \\)
> So \(b_u(k)\) is the total margin-relevant sensitivity located exactly \(k\) hops away from \(u\).
>
> **3) Per-node long-range profile and graph-level index.**
> We normalize the distance-binned sensitivities to obtain a per-node long-range fraction, then average over all nodes :
> \\( \rho_u^{(k)} = \frac{b_u(k)}{\sum_{s\in V} a_s}, \qquad \rho_G^{(k)} = \frac{1}{|V|}\sum_{u\in V}\rho_u^{(k)}. \\)
>
> Thus, \\(\rho_u^{(k)}\\) is the fraction of margin-relevant sensitivity at distance \(k\) hops from \(u\), and \\(\rho_G^{(k)}\\) averages this fraction over all nodes.
>
> Thank you again for this clarification request. We hope the above makes the computation clearer, and we will state the graph-level distance-binned definition more explicitly in the updated version.

---

> ### Comment · Reviewer_Btf2 · 2025-11-25
>
> I thank the authors for their clarification. After considering the definition more carefully, I am not convinced that the proposed measure captures long-range dependencies for graph-level tasks.
>
> The quantity $b_u^k$ (or $\rho_u^k$ ) effectively measures the total (or relative) contribution of k-hop neighbors to the margin. Since this quantity will naturally be large whenever a node has (relatively) many k-hop neighbors or when a non-central node (which appears often as a k-hop neighbour for large k) happens to be important for the classifier’s decision, it is not clear to me that it reflects long-range task dependence. Rather, it seems to capture structural properties of the graph (e.g., degree distribution, centrality patterns) rather than the intrinsic long-range nature of the task or model.
>
> This interpretation also appears consistent with the empirical results. In Table 1, graph-level tasks exhibit much higher $R^2$ for the Lasso regression, which would be expected if the measure correlates with structural signatures rather than long-range task difficulty. Similarly, Figure 3 suggests that PageRank alone achieves strong predictive performance, which is unsurprising given that PageRank captures centrality.
>
> While I understand the motivation behind the node-level definition, the graph-level definition seems qualitatively different, and the empirical behaviour of the two measures also differs substantially. This raises concerns about whether the proposed quantity consistently captures the intended notion of long-range dependencies across task types.

---

> > ### Author Response · Authors · 2025-11-26
> >
> > We thank the reviewer for the opportunity to clarify.
> >
> > While $b_u(k)$ (or $S_{u,g}(k)$) is an unnormalized mass at distance $k$, our diagnostic is the bounded ratio
> > $$
> > \rho_{u,g}=\frac{S_{u,g}(1)}{\sum_{k\ge 1} S_{u,g}(k)}\in[0,1].
> > $$
> > If a graph has many nodes at $k\ge 2$, this mainly increases the denominator and thus decreases $\rho$ unless the one-hop term truly dominates. Hence, $\rho$ is explicitly constructed to mitigate degree effects and to measure whether margin-relevant sensitivity is concentrated locally (high $\rho$) or spread to longer ranges (low $\rho$).
> >
> > Additionally, on TU datasets the degree dispersion is small, so variations in $k$-hop mass induced purely by degree heterogeneity are limited in this setting. Additionally, a degree-only baseline (min/mean/max degree) yields near-zero or negative test $R^2$, whereas richer structural indicators perform substantially better (See Table in Rebuttal Part 4).
> >
> > Finally, if a graph contains one (or several) highly central nodes, hop distances compress (small effective diameter), so information that would otherwise be "far" becomes reachable within $1$ or $2$ hops. In that regime it is normal that $\rho$ reports higher one-hop dominance: "long-range" is defined relative to the graph metric, and the graph itself makes the dependency effectively short-range in hop count.
> >
> > ---
> >
> > Thank you again for the constructive discussion.

---

### Author Response · Authors · 2025-11-30

We would like to sincerely thank all reviewers for the time and effort they dedicated to reading our work and engaging in the discussion; their comments helped us improve the paper. We have updated the PDF to address the questions, suggestions, and concerns raised. Below, we summarize the main additions and the corresponding issues resolved during the rebuttal.

- We clarify **the novelty of our Long Capture Index**, refine the definition of the **graph-level margin** in Section 3, and provide additional experiments on the behavior of our long-range index and its correlation with the classification margin (see Appendix A.2, Tables 2–3; reviewers Btf2 and REWK).


  &nbsp;


- To improve the interpretability of how the structural indicators used in rewiring methods relate to the **Long Capture Index**, we add degree-only and random baselines. These results indicate that **the structural descriptors leveraged by rewiring methods capture the aspects of graph topology most related to the model-derived $p_u$**, whereas degree alone is insufficient to explain the long-range sensitivity measured by our index (see Appendix A.2, Tables 4–5; reviewers Btf2, nToS, and REWK).

  &nbsp;


- We **extend our study to the Long-Range Graph Benchmark (LRGB)** in Appendix B (see Tables 9–11), comparing against rewiring methods, virtual-node methods, multi-hop approaches, and graph transformer models. Across all settings, **our method consistently improves the backbone GNN and reaches state-of-the-art performance**, while being **45–95% faster** than graph transformer variants (reviewers Btf2, XWMr, and REWK).

  &nbsp;


- To strengthen the attribution of **FLAN**’s improvements to mitigating long-range dependencies, we conduct a **series of ablation studies** and complementary analyses. We empirically show that **FLAN improves the margins of graphs that are most sensitive to long-range effects** (see Appendix C, Figures 5–6 and Table 12; reviewers Btf2, XWMr, and REWK).

  &nbsp;


- Finally, in Appendix C (Figures 8–11), **we analyze how margin-aligned sensitivity mass is distributed across hop distances and identify which hop ranges most drive FLAN’s gains** (reviewer REWK).

  &nbsp;

**We thank again the reviewers again** for their thoughtful reviews and constructive discussion, which significantly improved our paper.

---

**Given that the concerns raised pertained to clarity or to experimental aspects that we have now addressed through revisions and additional experiments (and not to the soundness, originality or importance of the work)  we hope the updated manuscript will be considered in that light, as the initial ratings may not fully reflect its current contributions.**

---

### Meta-Review · Area_Chair_KBU6 · 2026-01-04

**Summary:**

This paper investigates long-range interactions GNNs. It introduces a Jacobi-based metric that quantifies the contribution of single-hop interactions to decision margins, relative to interactions spanning two or more hops. The authors demonstrate that this metric can be roughly predicted from the graph structure alone. Finally, this paper proposes a rewiring-free layer designed to enhance long-range information propagation.

**Reviewer Concerns:**

The authors and reviewers engaged in fruitful communication, which effectively improved the quality of the paper. However, some questions remain unresolved.

1. Reviewer Btf2 not convinced that the proposed measure captures long-range dependencies for graph-level tasks. AC carefully read the definitions in the paper and the discussion between the two parties, and believes that the reviewers' concerns are reasonable, and the rationale for graph-level scores needs further clarification.
2. Reviewer nToS pointed out the inconsistent performance of FLAN on small heterophilous datasets, a point the authors agree with. Given the small size of the WebKB dataset and the reported flaws in the Squirrel/Chameleon dataset, AC believes a better rebuttal for the authors would be to provide experiments on larger heterophilous datasets (such as OGBN-Year). The limited results available so far diminish AC's confidence in the practical applicability of this paper.
3. Reviewer XWMr believes the results provided by the authors are insufficient to determine the validity of the proposed method. In particular, AC has run LRGB and obtained similar results to those reported in their paper. Therefore, AC agrees with reviewer XWMr's conclusion that the authors' results are underestimated.
4. Reviewer REWK raised many questions. After reading the entire discussion, AC considered [W2.4], [W3], and [W5] to be particularly important. In particular, for a flagship conference like ICLR, AC believed the rigor of the paper could be further improved.

**Reviewer Scores:**

4,6,2,4

---

### Decision · Program_Chairs · 2026-01-26

Reject